# Controlled release of growth factors using synthetic glycosaminoglycans in a modular macroporous scaffold for tissue regeneration

Z. Söderlund [1✉], A. Ibáñez-Fonseca [1], S. Hajizadeh[2], J. C. Rodríguez-Cabello [3], J. Liu[4], L. Ye [2], E. Tykesson[1,5], L. Elowsson[1,5] & G. Westergren-Thorsson[1]

Healthy regeneration of tissue relies on a well-orchestrated release of growth factors. Herein, we show the use of synthetic glycosaminoglycans for controlled binding and release of growth factors to induce a desired cellular response. First, we screened glycosaminoglycans with growth factors of interest to determine $k_{on}$ (association rate constant), $k_{off}$ (dissociation rate constant), and $K_d$ (equilibrium rate constant). As proof-of-concept, we functionalized an elastin-like recombinamer (ELR) hydrogel with a synthetic glycosaminoglycan and immobilized fibroblast growth factor 2 (FGF2), demonstrating that human umbilical vein endothelial cells cultured on top of ELR hydrogel differentiated into tube-like structures. Taking this concept further, we developed a tunable macroporous ELR cryogel material, containing a synthetic glycosaminoglycan and FGF2 that showed increased blood vessel formation and reduced immune response compared to control when implanted in a subcutaneous mouse model. These results demonstrated the possibility for specific release of desired growth factors in/from a modular 3D scaffold in vitro and in vivo.

[1] Lung Biology, Department of Experimental Medical Science, Lund University, Lund, Sweden. [2] Division Pure and Applied Biochemistry, Lund University, Lund, Sweden. [3] BIOFORGE Lab, CIBER-BBN, Universidad de Valladolid, Valladolid, Spain. [4] Division of Chemical Biology and Medicinal Chemistry, Eshelman School of Phamarcy, University of North Carolina, Chapel Hill, NC, USA. [5] These authors contributed equally: E. Tykesson, L. Elowsson. ✉email: zackarias.soderlund@med.lu.se

Regeneration of tissue is a complex process involving multiple pathways, which need to be activated in a correct temporal and spatial manner. A plethora of growth factors and chemokines have been explored and mapped in vitro to gain a deeper understanding of their cellular effects, such as blood vessel differentiation with fibroblast growth factor 2 (FGF2), vascular endothelial growth factor (VEGF) and platelet-derived growth factor (PDGF)[1], bone formation with bone morphogenetic protein (BMP)[2,3] and proliferation of nerve cells with nerve growth factor (NGF)[4].

However, growth factors do not always translate in vivo. This is partially due to the potency of growth factors, which are quickly neutralized when released in vivo to minimize adverse effects[5,6]. To circumvent too quick neutralization, growth factors can be incorporated into scaffolds such as hydrogels, where release is limited due to low area to volume ratio, although this could limit cell movement. In contrast, a number of organs have interconnected and porous extracellular matrix (ECM) structures, such as lung[7], muscles[8], and bone[9], with exponentially higher surface to volume ratios. Multiple methods exist to create macroporous structures, cryogels being one of the most used, where the material forms around ice crystals. When the ice is removed, macroporous structures with characteristics of shape-memory properties are formed, which enables for injectability through a syringe[10]. The disadvantage of manufactured macroporous scaffolds incorporated with growth factors is quick release due to high exposure area.

To circumvent the fast release from macroporous structures, different strategies have been designed[11]; using aptamers[12], or creating a binding tag[12,13]. However, these approaches only allow for either binding of one single factor or only the loaded factors and not binding factors produced in the body. In nature, creating growth factor gradients, a fundamental step in the orchestration of cell development and repair[14,15], is to a great extent solved by glycosaminoglycans, which consist of linear carbohydrate motifs known to bind a plethora of growth factors and cytokines. Creation of glycosaminoglycans is template free and dependent on over 40 enzyme concentrations, and their mechanisms are not completely understood. This, in combination with the lack of sequencing technologies, have led to the use of long undefined glycosaminoglycans purified from different origins[16–20] that bind growth factors, thus with unknown sugar length and sulfation sequence. Moreover, batch effect and purity of animal-derived glycosaminoglycans have unfortunately led to fatal consequences when used in the clinic[21]. Recently, synthetic glycosaminoglycans have been developed[22], where carbohydrates are enzymatically added to a core, one by one. They are structurally defined with the possibility of either having an active group or a fluorochrome at the end of the core. The main difference compared to native glycosaminoglycans is that the engineered glycosaminoglycans are shorter. Sometimes, but not always, this is a disadvantage depending on the prerequisites for receptor binding, which may demand for longer chains[23].

Glycosaminoglycans in the body are part of the ECM. Numerous molecules, the most common being collagen, gelatin, and alginate, have been used in tissue engineering to mimic the dynamics of native ECM scaffolds[24]. The main limitation with these ECM molecules is that they are often purified from animals and algae, with the risk of being contaminated and hard to modify. Subsequently, a lot of effort has gone into making recombinant biomaterials e.g., elastin-like recombinamers (ELRs)[25], a repeated human elastin sequence expressed in bacteria that has already shown biocompatibility and improvement of tissue regeneration[25–27]. Due to the recombinant expression, several modifications can be made, such as introducing an arginyl-glycyl-aspartic acid sequence (RGD motif) for attachment, a matrix metalloproteinase domain for controlled degradation or functionalization of lysine groups for further modification.

In this study, we hypothesized that synthetic glycosaminoglycans could be used to functionalize a macroporous ELR scaffold for a tonic release of growth factors, creating a potent and localized growth factor gradient for the recruitment of cells such as immune and vascular cells, to initialize regeneration of tissue in situ.

## Results

### Screening of synthetic glycosaminoglycans for growth factor binding

To demonstrate the possibility to use synthetic glycosaminoglycans as a tool for customizing release rate of growth factors, a range of growth factors were screened on a microarray functionalized with 52 different synthetic glycosaminoglycans, referred to as 2–53 while 1 is commercially available heparin and, when applicable, the background value (0) is given (see Supplementary Data 1). Figure 1A–C shows binding of VEGF and hepatocyte growth factor (HGF) to glycosaminoglycans, furthermore, BMP4, CXCL12, IL-6, FGF2, PDGF-AA, KGF, and TGF beta 1 were also tested and data can be found in Supplementary Fig. 1 and Supplementary Data 2–10). Growth factors added to the microarray revealed three groups of glycosaminoglycans: those that showed high binding to all growth factors, those that showed low/no binding, and finally those that showed selective binding depending on the growth factor. To further study this on a kinetic level, we evaluated a subset of the glycosaminoglycans using surface plasmon resonance (SPR). One glycosaminoglycan with low to no binding to any of the growth factors, GAG nr 3; one high binding to all growth factors (except for TGF beta 1), GAG nr 19; and four with binding to only a subset of the growth factors, GAG nr 10, 26, 34 and 43. In contrast to microarray data, SPR enables label-free measurements of both association ($k_{on}$) and dissociation ($k_{off}$) rate constants, as well as the calculation of equilibrium rate constant ($K_d$). A few different patterns were observed in the SPR data of the analyzed glycosaminoglycans (Fig. 1E) (the raw data can be found in Supplementary Data 11). The synthetic glycosaminoglycan with the longest carbohydrate chain as well as the highest degree of sulphation, GAG No. 19 (highlighted in orange in Fig. 1E), was the only one to show binding to transforming growth factor beta 1 (TGF beta 1), keratinocyte growth factor (KGF) and connective tissue growth factor (CCN2). Additionally, GAG nr 19 generally showed the slowest release of growth factors (low $k_{off}$), but not always the highest $K_d$ between the glycosaminoglycans. Surprisingly, the short and non-sulfated GAG nr 3, which was chosen because of its low to no binding in the microarray data, showed binding to multiple growth factors. However, it had a faster release rate (higher $k_{off}$ values) compared to the other evaluated glycosaminoglycans, possibly causing a release of the growth factors during the washing step in the microarray experiment, resulting in low to no measurable binding. GAG nr 19 was chosen for further experiments due to its binding to both VEGF ($K_d = 137$ nM and $k_{off} = 3 \times 10^{-2}$ s$^{-1}$) and basic fibroblast growth factor (FGF2) ($K_d = 52$ nM and $k_{off} = 2 \times 10^{-4}$ s$^{-1}$).

### Functionalization of ELR hydrogel using synthetic glycosaminoglycans

To show the advantages of using a fully defined system as well as the efficacy of releasing growth factors via synthetic glycosaminoglycans, we set up an endothelial tube formation assay based on a previous work[28]. In this case, human umbilical vein endothelial cells (HUVECs) were cultured on an ELR hydrogel containing synthetic glycosaminoglycans, GAG nr 19, instead of the normally used Matrigel hydrogel. One main

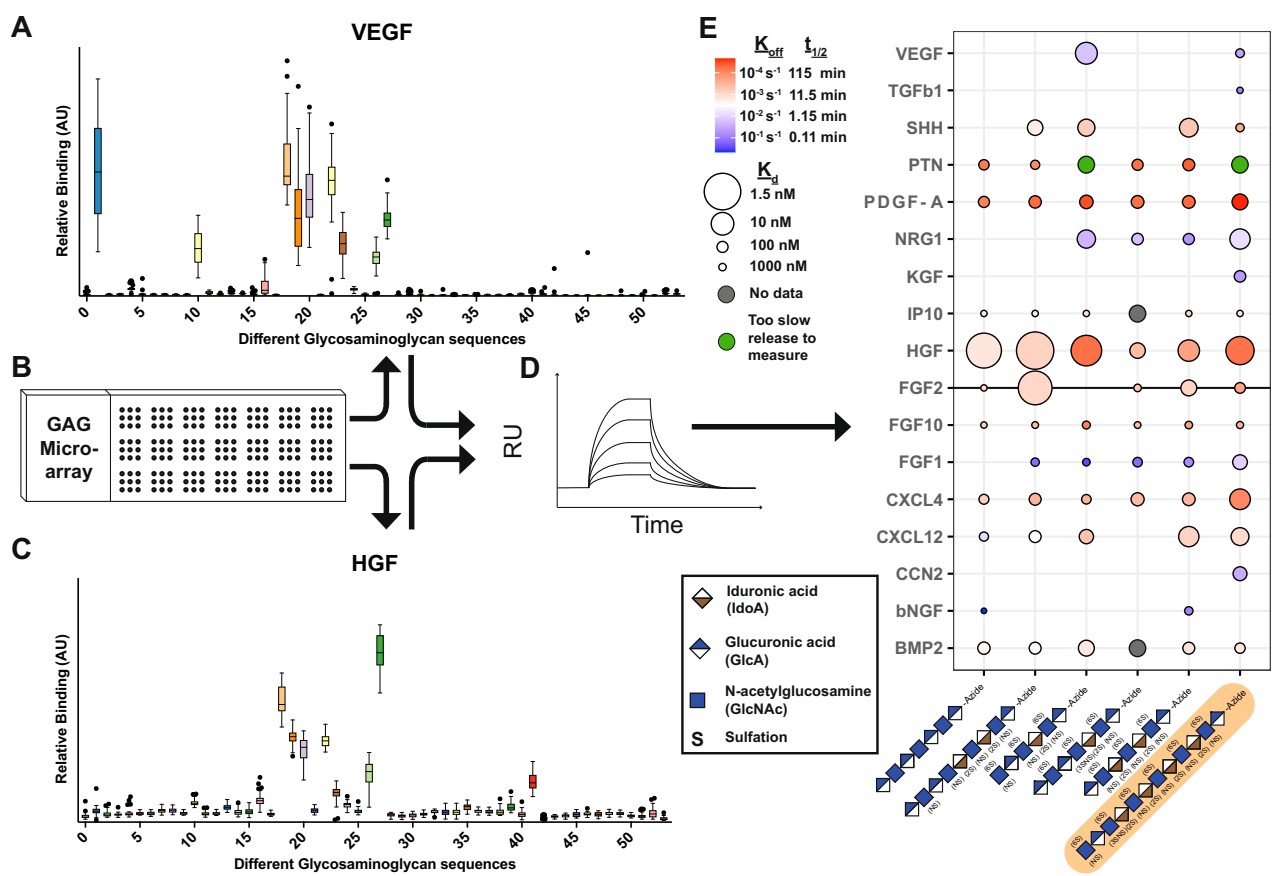

**Fig. 1 Screening of growth factor interactions with synthetic glycosaminoglycans. A** Relative binding of VEGF to 52 different synthetic glycosaminoglycans measured on a microarray. **B** An illustration of the microarray. **C** Relative binding of HGF in the microarray to show the difference in binding pattern depending on the glycosaminoglycan. For **A** and **C** data is shown in a box and whisker (as Tukey) of 36 binding spots for each synthetic glycosaminoglycan. **D** Six synthetic glycosaminoglycans with different binding patterns were chosen for further analyses using surface plasmon resonance (SPR) to determine their binding in a quantitative way compared to the semi quantitative analysis that the microarray allows. **E** Heatmap showing the binding of six selected synthetic glycosaminoglycans on the *x*-axis and the tested growth factors on the *y*-axis. The glycosaminoglycans are ordered in increasing degree of sulfation, GAG nr 3, 43, 34, 10, 26, and 19. The color shows the $k_{off}$ value, where red illustrates a slow release while a blue color a fast release. The size of the circle illustrates the equilibrium dissociation constant, $K_d$, where a larger circle is for higher binding while a smaller circle is for lower binding. There are also two points with no data indicated in gray and two points that did not show any release at all, indicated in green. For convenience, we have highlighted the growth factor FGF2 (gray line) and GAG nr 19 used in later experiments (orange highlight). Factors tested that did not show any interactions in the SPR analysis were: IL- 6, IL-11, TGF beta 2, TGF beta 3, CCL2, CCL3, CCL4, CCL23, Wnt2, G-CSF, NOV, EGF, IGF1, GLP, SCF, CXCL2, CXCL7.

advantage of using an ELR hydrogel is that it is completely defined and is only made of an elastin-derived recombinant protein, while Matrigel contains both glycosaminoglycans and growth factors that would therefore interfere with the experiment. The results (Fig. 2) showed no differentiation in the negative control (only medium) (Fig. 2B), nor with only FGF2 (Fig. 2D) or synthetic glycosaminoglycans (Fig. 2E) after 24 h, characterized by cells forming clusters with no tube formation between them. In contrast, numerous tube formations were seen for HUVECs cultured on ELR hydrogel with immobilized glycosaminoglycans, GAG nr 19, and bound FGF2 (Fig. 2F). In addition, cells arranged in loops indicating full differentiation, similar to what was seen in the positive control (Fig. 2C), where cells had been stimulated with complete medium which is supplemented with both glycosaminoglycans (heparin) and FGF2.

**Production of a macroporous elastin-like scaffold**. There is a multitude of tissues where the ECM is organized into a macroporous structure e.g., bone, lung, muscle; however, incorporation of growth factors in a manufactured macroporous structure may result in low retention of growth factors as the surface area is

exponentially larger. We hypothesized to overcome this issue by combining synthetic glycosaminoglycans with an ELR able to form macroporous structures, to mimic the in vivo situation where growth factor gradients are largely controlled via glycosaminoglycans[15]. To prove that ELR could be used to provide a scaffold for a multitude of tissues with different ECM structures, we successfully used ELR in a cryogelation method that involved the use of ice crystals of defined sizes during biomaterial polymerization (Fig. 3C–E). In comparison to ELR hydrogels, the ELR cryogels showed a large structural difference by scanning electron microscopy (SEM), where the cryogel showed an interconnective macroporous structure while the hydrogel was non-porous. In the cryogel, the size of the formed pores closely resembled the size of the ice crystals added during production, opening the possibility to customize both the desired size and ratios between different pore sizes, and did not change with the addition of synthetic glycosaminoglycans as can be seen in Supplementary Fig. 2A, B.

The mechanical properties of ELR cryogels with three different pore sizes, namely 100–200, 200–500, and 500–1000 μm, were studied by oscillatory rheology and uniaxial compression testing.

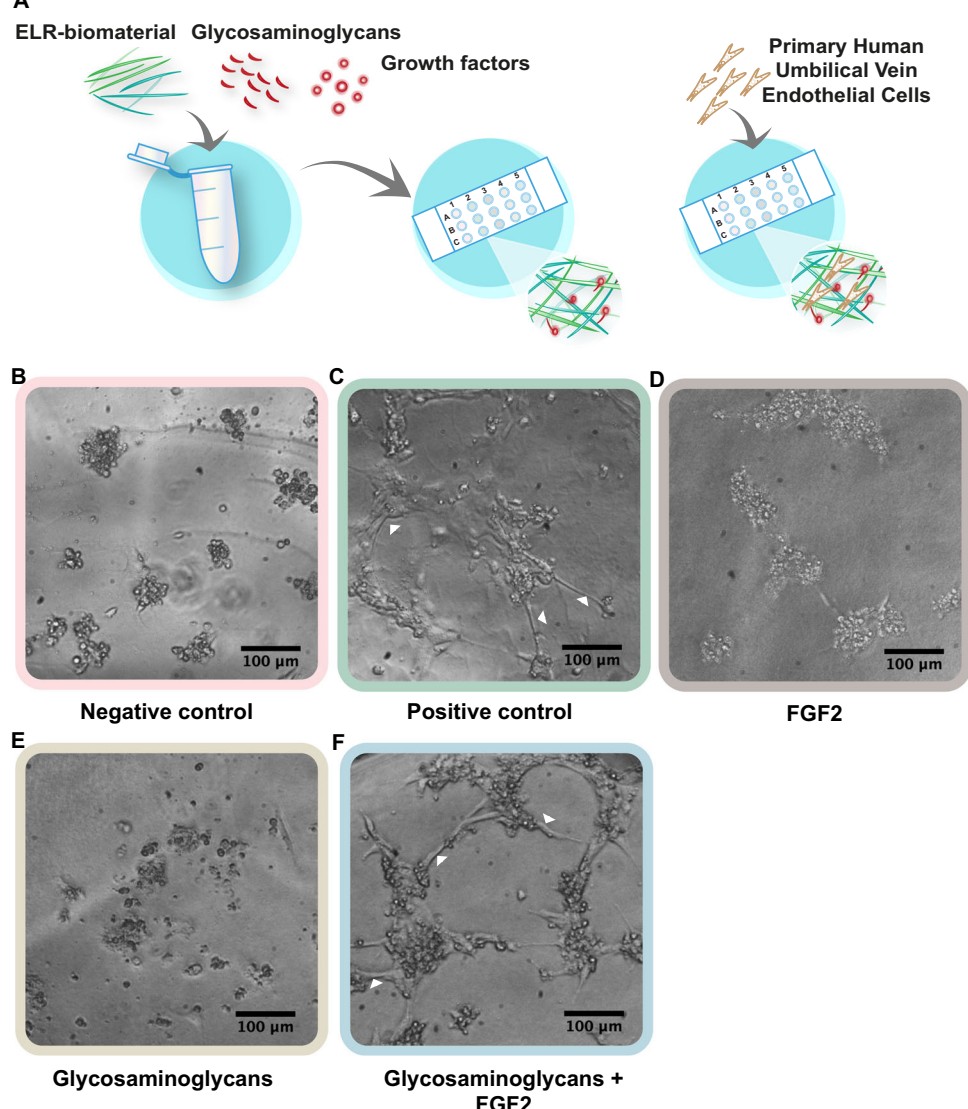

**Fig. 2 Differentiation of HUVECs using synthetic glycosaminoglycans. A** Schematic representation of the mixture of the synthetic glycosaminoglycans and growth factors (FGF2) with an elastin-like recombinamer (ELR) to give functionalized hydrogels. Synthetic glycosaminoglycans were conjugated to the ELR through 'click chemistry'. HUVEC cells were added on top and allowed to differentiate for 24 h. **B** Undifferentiated HUVECs when cultured in medium without supplements, characterized by cells clumping together and dying. **C** Differentiated HUVECs when stimulated with complete medium, characterized by the formation of tube-like structures (highlighted with white arrows). **D, E** Undifferentiated cells when stimulated with only FGF2 or only synthetic glycosaminoglycans. **F** Differentiated cells when stimulated with the combination of synthetic glycosaminoglycans and FGF2, (tube-like structures highlighted with white arrows).

Specifically, oscillatory rheology results showed no significant differences between the groups with an average elastic modulus ($G'$) of 115 Pa (Fig. 3F). This was in line with the compression tests, which resulted in similar tangential Young's elastic moduli ($E$) for the different groups (Fig. 3G), and no cryogel failure or fracture was observed in any of the tests for any sample, with all of them resembling highly elastic tissue. Both results highlight the possibility of changing the microstructure without affecting the mechanical properties of the scaffolds within the tested range. Interestingly, this allows the evaluation of the sole effect of various pore sizes in cell culture or during cell invasion upon implantation of the scaffolds in vivo, without modifying biomaterial chemistry or scaffold mechanics. On the other hand, the ELR hydrogel gave a tangential $E$ value of $3077.4 \pm 98.6$ Pa (Supplementary Fig. 3), while the $G'$ was found to be $633.8 \pm 95.9$ Pa (see Supplementary Fig. 4 for the strain sweep

rheology curve). Both measurements resulted in a much higher stiffness than for all the groups of the cryogel counterparts.

**Tuning the in vivo response using synthetic glycosaminoglycans for a tonic release of growth factors**. To evaluate the effect of having a tonic release of growth factors from a macroporous structure, an in vivo study was carried out with cryogels having a pore size of 200–500 µm, chosen as an example as it resembles the distal lung ECM. The ELR cryogels were functionalized with the synthetic glycosaminoglycan, GAG nr 19, that showed the slowest release ($k_{off}$) for FGF2. Which were subsequently incubated with FGF2 and implanted subcutaneously in mice (Fig. 4A). Prior to this, a pilot study was performed to assess the timespan of the experiment with a focus on the degradation of ELR cryogels, to compare with the results from a previous in vivo study with ELR hydrogels[25]. Cryogels were labeled with the 680RD fluorescent

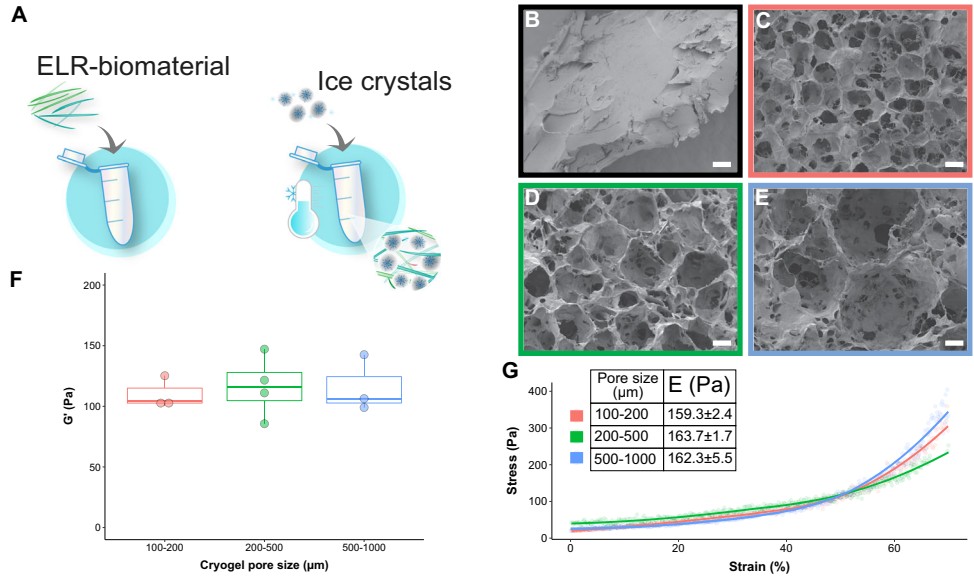

**Fig. 3 The creation of a macroporous biomaterial that takes advantage of synthetic glycosaminoglycan binding properties mimicking non-solid organs.**
**A** Cryogel formation: Ice crystals were mixed during the formation of the biomaterial, resulting in the ELR forming a network around the ice crystals. The ice was removed upon freeze-drying and a macroporous structure remained. SEM pictures showing the structure of the materials, **B** (black outline) corresponds to the ELR hydrogel, **C** (red outline) to the cryogel made with ice crystals of 100–200 μm in size, **D** (green outline) to the cryogel with 200–500 μm in size and **E** (blue outline) to the cryogel with 500–1000 μm in size. Scale bar is 100 μm. **F** Rheological measurements of cryogels with different pore sizes showing similar storage moduli ($G'$) with an average of 115 Pa. **G** Mechanical testing: Stress–strain curves were obtained with compressive tests of the cryogels with different pore sizes, i.e., 100–200 μm (red), 200–500 μm (green) and 500–1000 μm (blue). The embedded table shows the tangential Young's elastic modulus ($E$) for each group, showing no significant differences. **F, G** data are represented as mean ± SD; $n = 3$ for all the groups, except for the rheology tests 200–500 μm group ($n = 4$).

infrared dye at different concentrations to determine the optimal dilution. The highest brightness was observed for 2 μg dye per 50 mg cryogel (100 times dilution) (Supplementary Fig. 5A) and was further used for cryogels that were implanted subcutaneously. The labeled cryogels were monitored at 1, 4, 6, 7, 10, and 14 days with no difference in size and intensity observed (Supplementary Fig. 5B, C) and no detectable cryogel elsewhere (Fig. 4B as well as Supplementary Fig. 5C). Thus, to see a larger difference between samples, 8 weeks was selected as end point for the main experiment. After this time, blood vessel formation was evaluated by calculating the number of CD31 positive cells in the cryogel normalized by area (Fig. 4C, G, H, Negative control staining in Supplementary Fig. 6E). An increase in blood vessel formation was seen in the samples containing FGF2 compared to control (0.0061 μm$^{-2}$ compared to 0.004 μm$^{-2}$), and an additional small increase was seen with the addition of synthetic glycosaminoglycans (0.0064 μm$^{-2}$). Red blood cells were seen inside the blood vessel confirming that they were connected to the main circulation (red arrows Fig. 4I, J). With further histological evaluation of the samples with a hematoxylin and eosin staining (H&E) (Fig. 4I, J), we discovered a difference in the presence of giant multinucleated cells, which is a common feature of a foreign body response[29]. No consistent difference in capsule size was seen, Supplementary Fig. 6I. After binary scoring of the samples for the presence and absence of giant cells (Fig. 4D), we found fewer positive samples in the FGF2 group (40%) and the least positive samples in the combined synthetic glycosaminoglycans and FGF2 group (33.3%) in comparison with control (100%). To confirm these findings, a multiplex ELISA was performed with the plasma collected at the end point. For IL-6, VEGF, FGF2, MMP8, and TIMP-1 there were no observed differences (Supplementary Data 12 and Supplementary Fig. 7). Though, interestingly, for IL-4 (Fig. 4E), there was a decrease between control samples and FGF2 (199.9 pg/ml compared to 182.0 pg/ml) and an

additional decrease for the combination of synthetic glycosaminoglycan and FGF2 (166.8 pg/ml). IL-4 is both linked to M2 macrophages as well as to the formation of giant multinucleated cells[30,31], which led us to evaluate the immune response caused by the different groups by calculating the relation between M1 and M2 macrophages by taking M2 over M1 + M2. Positive M1 and M2 cells were counted using QuPath. No positive cells were found in the negative controls (Supplementary Fig. 6H, negative control staining in Supplementary Fig. 6F, G). A trend could be observed with an increase in the number of M2 macrophages with the addition of FGF2 compared to control (29.2% compared to 15.6%), and with a further increase in the combined synthetic glycosaminoglycan and FGF2 samples (44.4%) (Fig. 4F, K, L). Taken together, tuning the in vivo effect by tonic release of FGF2 can be achieved using synthetic glycosaminoglycans. VEGF was also evaluated, but no additional effect was seen (Supplementary Fig. 6A–D). All collected data can be found in Supplementary Data 12 and Supplementary Fig. 7, including data for weeks 2, 4, and 8. In Supplementary Figs. 8–11, pictures of implanted ELR cryogels for all groups for each staining at each timepoint are found.

## Discussion
In this work, we show that synthetic glycosaminoglycans bind to a wide variety of growth factors at different $K_d$, $k_{on}$, and $k_{off}$. Moreover, we show that these glycosaminoglycans can be combined with a biomaterial such as the ELR for tissue engineering and regenerative medicine (TERM) applications.

Previous work have combined hydrogels with heparin[32] or with disaccharides[33], but few so far has reported about the characterization of synthetic glycosaminoglycans and their possible use in release of growth factors. Herein, we show that synthetic glycosaminoglycans have binding strengths comparable to heparin, despite being shorter in lengths, and with the advantage

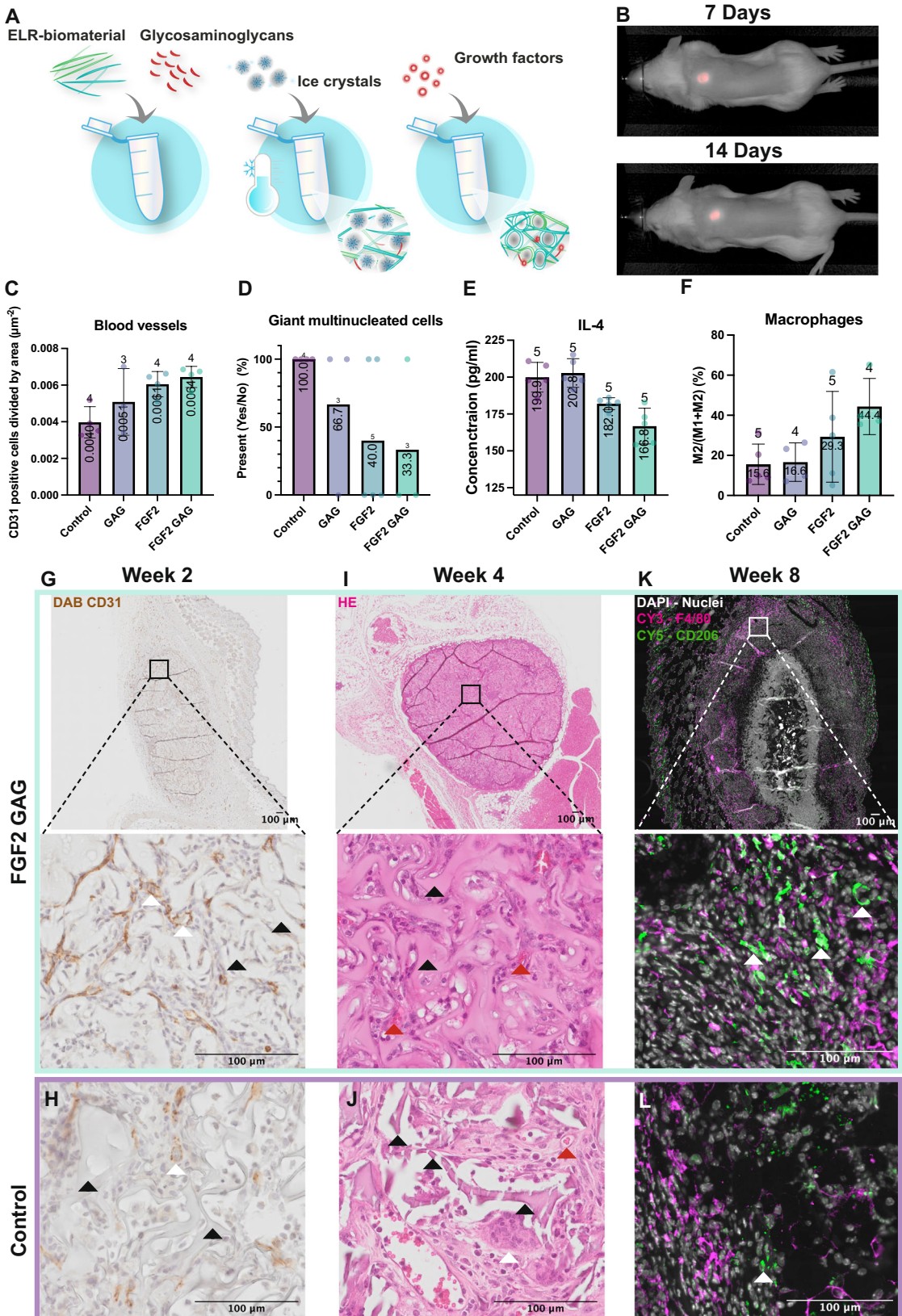

of having a chemically defined structure, which previously only has been possible for disaccharides.

We show possible applications for the synthetic glycosaminoglycans with hydrogels as well as in macroporous structures, opening up the possibility to engineer a desired cellular response both in vitro and in vivo. The developed toolset is relevant for a large variety of organs when studying tissue regeneration due to its ease of use and modular design.

Glycosaminoglycans are important constituents in tissue repair[34,35], for guiding cellular activity and retaining and

**Fig. 4 Testing synthetic glycosaminoglycans as mediators of tonic release of growth factors in vivo for their effect on blood vessel formation and the immune response. A** The ELR material was mixed with the synthetic glycosaminoglycans to form a covalent bond. Then ice crystals were added, and the biomaterial was formed around the ice, creating a macroporous structure. Lastly, the growth factor was added, which bound to the synthetic glycosaminoglycan. **B** Images from the pilot experiment show no change in size of the fluorochrome-labeled ELR after 7 and 14 days and no accumulation elsewhere in the body. **C** The graph shows the number of positive CD31 cells divided by the area after 8 weeks showing the highest number of blood vessels in the combined group of synthetic glycosaminoglycan and FGF2. There was a limited number of slides that could be recovered from each biological sample and therefore the number above each bar shows how many biological samples were analyzed. All error bars are SD in the figure. **D** H&E-stained samples were scored for the presence (1) or absence (0) of multinucleated cells after 8 weeks. **E** IL-4 was measured in plasma after 8 weeks. **F** The M2 to total number of macrophages by staining one slide for M1 and one for M2 macrophages, which were then counted and normalized to total cell number, dividing M2 with M1 + M2 at 8 weeks. **G** 2 weeks after implantation showing the CD31 DAB staining for the samples with FGF2 and synthetic glycosaminoglycans, **H** shows the same for the control biomaterial. **I** 4 weeks after implantation showing the H&E staining for the group with synthetic glycosaminoglycan and FGF2 where the black arrows point at the ELR material and the red arrows point at blood vessels with red blood cells inside. **J** Shows the same for the control biomaterial where the arrows point at the multinucleated cells, the black arrows point at the ELR material, which demonstrates how the biomaterial has started to degrade by breaking into pieces. Red arrows point at blood vessels with red blood cells inside. **K** 8 weeks after implantation showing the M2 staining with DAPI in white, F4/80 in magenta, and CD206 in green. The white arrows highlight the CD206 positive cells. **L** Shows the same for the control biomaterial where less CD206 positive cells can be seen in green.

releasing growth factors and have thus been used extensively in regenerative medicine[32,36,37]. But until recently, the use of glycosaminoglycans in therapy has been limited to extracted fractions from animal tissues or cell extracts, where only the overall sulfation can be defined but not the sequence, causing problems with standardization[38]. Synthetic glycosaminoglycans extend the ability to study and use glycosaminoglycans in demanding applications where batch differences cannot be tolerated, which is inevitable when using a purified product of animal origin[39], as one can move away from defining glycosaminoglycans by its percentage of sulfation and di-saccharides and instead use a well-defined glycosaminoglycan sequence. It is known that length and sulphation pattern of glycosaminoglycans affect binding capacity of growth factors. The microarray with glycosaminoglycans was used to identify synthetic glycosaminoglycans with large difference in binding and in sulfation pattern, which was further analyzed using SPR. The main advantages with SPR are that it does not require washing steps, nor the use of antibodies. Furthermore, the technique is not dependent on knowing how much the glycosaminoglycans is attached to the surface, which is a prerequisite for microarrays. Using SPR revealed how diverse glycosaminoglycans bind and release growth factors as both $k_{on}$ (association rate constant) and $k_{off}$ (dissociation rate constant) can be measured. For example, even glycosaminoglycans with no sulfation, showing little or non-binding on the microarray screening, had a $k_{on}$ and a $k_{off}$ that could be measured for multiple growth factors, although showing an overall lower binding with the lowest $k_{off}$ value for each growth factor. Most binding studies for glycosaminoglycans are done using disaccharides, while studies of glycosaminoglycans in the range of 6 to 12 sugars are limited. Linhardt et al have studied the interaction between FGF2, FGF7, FGF10, TGF beta 1, HGF and longer glycosaminoglycans where they removed specific sulfation[40]. Interestingly, they saw that N sulfation was the most important alteration for binding FGF2. This is in line with our results where we saw better binding to GAG nr 43 with high percentage of N sulfation than to GAG nr 19 with lower percentage of N sulfation. In contrast to Linhardts study we used synthetic glycosaminoglycans which might explain some discrepancies. There was a large heterogeneity between the lowest and highest $K_d$ for the synthetic glycosaminoglycan as $K_d$ differed up to 1000 times (between GAG nr 43 and nr 3 for FGF2), which proves that there is a possibility to fine tune GAG-growth factor binding to obtain specific cellular outcomes.

The critique of synthetic glycosaminoglycans has been that they have a shorter length compared to native glycosaminoglycans, but herein we show that short synthetic glycosaminoglycans

bind several growth factors. Moreover, we wanted to investigate if synthetic glycosaminoglycans can act as co-receptors as well as creating growth factor gradients and steer cell response. To test this, we set up an in vitro differentiation assay for HUVECs, which often are cultured in Matrigel. The disadvantage with Matrigel is that it is a cell extract with large batch to batch variations in the content of growth factors and glycosaminoglycans already present in the hydrogel. Instead, we have used the recombinantly expressed ELR making use of the alkyne activated part of the ELR to "click in"[41] the synthetic glycosaminoglycans, as these have an azide end that create a covalent bond with alkyne when mixed, thus resulting in an ELR surface functionalized with synthetic glycosaminoglycans. The experiment showed that the bound synthetic glycosaminoglycans were able to interact with FGF2 and that this complex in turn activated the differentiation of HUVECs, by acting as co-receptor, to form tubular structures between cells.

FGF2 is interesting in the regeneration field as one of the main inducers of vascularization together with VEGF[42], as vascularization is key to replenish the site with oxygen and nutrients and is often the first hurdle to overcome for a successful graft[43]. This well-defined model system, using a biomaterial with a known elastin sequence in combination with synthetic glycosaminoglycans, opens for the possibility to test for inhibitors and activators of vascularization that otherwise might already be present in undefined ECM hydrogels such as Matrigel[6].

In recent years, the importance of the microenvironment has received increasing recognition, not least in the TERM field. As most human tissues have a porous structure, we aimed at developing a system where we could combine slow and controlled release of growth factors and simultaneously mimicking the macroporous structure. Making use of the unique properties of the ELR we could form an interconnected macroporous structure with controlled pore size and with synthetic glycosaminoglycans. Combining the cryogelation technique by including ice crystals of known size during manufacturing, allowed for the development of tunable 3D structures. This method produced an interconnected macroporous structure with large potential for customizing desired pore size (in the range of 100–1000 μm) without the need to add anything else than water and ethanol that are easily removed afterwards. As previously demonstrated[44], pore size does not correlate to stiffness within the tested range and can thus be independently tuned to match the application. In the case of the hydrogels, their much more compact and dense structure gave a higher stiffness in comparison with the cryogels, as expected, but still in the range of soft tissues. Given that the glycosaminoglycans used in the bioconjugation will bind

approximately 1/500 of the total binding sites available for the crosslinking, their presence will have a residual effect on the hydrogel formation process and, therefore, on its structure and mechanical properties. In addition, $E$ and $G'$ values obtained by uniaxial compression tests and shear rheology, respectively, gave slightly different results due to the inherent differences of the two methods used to characterize the mechanical properties of the hydro- and cryogels. While we could have chosen to use only one of them, we aimed at giving as much data as possible to allow for the comparison with other studies, where authors show either $E$ or $G'$ for their constructs. The mechanical data showed a physiological stiffness resembling softer organs[45] and a low elastic modulus avoiding myofibroblast activation in response to stiff substrates, as previously suggested[46,47]. Moreover, the developed cryogels are not intended to act as tissue substitutes, but as temporary cell-instructive scaffolds[48] that promote in situ tissue regeneration by releasing exogenous and capturing endogenous growth factors through glycosaminoglycan binding, and by providing a niche environment for cells driving regeneration.

By combining our two systems, we wanted to investigate its utilization in vivo. Our initial plan was to use synthetic glycosaminoglycans with both VEGF and FGF2 binding, as these growth factors have been shown to act synergistically, where VEGF induces PDGF expression and FGF2 induces PDGF receptor expression, resulting in the maturation of blood vessels[42]. However, we saw no added effect of including VEGF in the macroporous ELR, when combining it with synthetic glycosaminoglycans. We suggest that this was due to the 100 times faster release ($k_{off}$) of VEGF compared to FGF2 for the selected synthetic glycosaminoglycan, but could also be due to the 2.5 times higher equilibrium dissociation constant ($K_d$). We believe this indicates that focus should be on $k_{off}$ and not on $K_d$, when looking to sustain the efficacy of growth factors when using synthetic glycosaminoglycans. As none of the synthetic glycosaminoglycans had the exact same $K_d$ and had a large difference in $k_{off}$, further screening is needed to determine the importance of $K_d$ and $k_{off}$.

As previously discussed, blood vessel formation is one of the critical points in tissue regeneration and therefore this was our main readout from the in vivo study. Interestingly, we did see an increase in blood vessel formation when synthetic glycosaminoglycans and FGF2 was combined in the cryogel compared to the control ELR cryogel. This highlights the benefit of using synthetic glycosaminoglycans in regenerative medicine to increase the local effect of growth factors. Intriguingly, we found a difference between the groups for the occurrence of giant multinucleated cells, a common foreign body response to a subcutaneous implant[49]. In samples where the combination of synthetic glycosaminoglycan and FGF2 had been used, only one was positive for the presence of giant multinucleated cells, while the other groups had several positive samples, and for the group with control ELR cryogels all were positive. This implied that a slow release of FGF2 via the synthetic glycosaminoglycan affected the immune response to the implant, enhancing the integration rather than rejecting the material. To further investigate the change in immune response, we looked at a panel of chemokines and growth factors in plasma. Our results showed a decrease in IL-4 levels between the FGF2 group and the group with both FGF2 and synthetic glycosaminoglycans. Too high levels of IL-4 are linked to multinucleated cells, and in moderate levels, linked to M2 macrophages[30,31]. There are two main macrophage phenotypes, M1 and M2, where high levels of M1 macrophages characterize a more pro-inflammatory and rejection response, while a high levels of M2 macrophages characterize wound healing and integration of the implant[50]. The higher ratio of M2 over M1 macrophages for the FGF2 and glycosaminoglycan group further supported the shift in immune

response. Our results indicate that slow release of FGF2 from glycosaminoglycans affects immune response, as we did not see any change with the addition of only synthetic glycosaminoglycans. For the in vivo studies, only one pore size was tested (200–500 μm), warranting further studies to evaluate other sizes, as larger pores have been shown to be more anti-inflammatory[44]. Furthermore, the bulk release of growth factors from the cryogels should be evaluated, as growth factors may jump from one glycosaminoglycan to another, which may result in a longer retention time in the cryogel. Additionally, the binding of growth factors should be tested in regard to the volume of cryogels as well as the amount of glycosaminoglycans in order to see how these factors contribute.

Taken together, our results indicate that the synthetic glycosaminoglycan plays an important role in the sequestration of growth factors, resulting in slow release of growth factors in vivo that in turn steers cell response to induce blood vessel formation and to modulate the systemic and local immune response.

In conclusion, we show that synthetic glycosaminoglycans are promising in steering cell fate both in vitro and in vivo when developing complex biomaterials. The system allows for the design of unique kinetics for binding and releasing growth factors to steer cellular responses. Moreover, we provide a complete workflow for creating complex macroporous ELR structures, which are easily customizable and functionalized via 'click chemistry'. Finally, we demonstrate that synthetic glycosaminoglycans, with the advantage of being structurally defined, retain their biological functions and thus have potential to become a cornerstone in regenerative medicine.

## Method

**Glycosaminoglycan microarray screening.** The construction of synthetic glycosaminoglycans has been previously described[22,51]. In short, the glycosaminoglycans were chemoenzymatically produced by adding one sugar molecule at a time with purification in between. The synthetic glycosaminoglycan was built on an azide backbone, which was printed on a cyclic alkyne microscope slide using a pipetting robot. For the screening, the proteins; HGF (294-HG-025/CF, RnD systems), VEGF (293-VE-010/CF, RnD systems), BMP4 (314-BP-010/CF, RnD systems) CXCL12, (350-NS-010/CF, RnD systems), IL-6 (7270-IL-025/CF, RnD systems), FGF2 (ab9596, Abcam), PDGF-AA (221-AA-010, RnD systems), KGF (251-KG-010/CF, RnD systems), and TGF beta 1 (7754-BH-025/CF, RnD systems) were dissolved at a concentration of 10 μg/ml in PBS (1891014, Thermo Fisher), Tween 20 (P1379, Sigma-Aldrich,) and 10% BSA (A2153, Sigma-Aldrich), and 100 μl of the solution was added to the slide and incubated in a humified chamber for 60 min. The slides were rinsed in PBST with 1% BSA for 3 min and then in deionized water for 3 min. Antibodies towards the protein were dissolved in PBST 10% BSA according to the manufacturer's recommended concentration, added to the slide and incubated in a humified chamber for 60 min (HGF Ab, MAB694, 0.4 μg/ml, Mouse anti Human, RnD systems) (VEGF Ab, MAB293, 0.4 μg/ml, Mouse anti Human, RnD systems) (BMP4 Ab, MAB7571, 0.4 μg/ml, Mouse anti Human, RnD systems) (CXCL12 Ab, MAB350, 0.4 μg/ml, Mouse ant Human, RnD Systems) (IL-6 Ab, MAB2063, 0.4 μg/ml, Mouse ant Human, RnD systems) (FGF2 Ab, ab92337, 0.5 μg/ml, Rabbit anti Human, Abcam) (PDGF-AA Ab, MAB221, 0.4 μg/ml, Mouse anti Human, RnD systems) (KGF Ab, MAB251, 0.4 μg/ml, Mouse anti Human, RnD systems) (TGF beta 1 Ab, MAB240, 0.4 μg/ml, Mouse anti Human, RnD systems). The slide was washed with PBST 1% BSA and deionized water for 3 min. A secondary antibody conjugated to the fluorochrome Alexa fluor Plus 647 (IgG Alexa 647 plus, A32728, 1/1000, Goat anti Mouse, Thermo Fisher or IgG Alexa 647 plus, A32733, 1/1000, Goat anti Rabbit, Thermo Fisher) was added at a 1/1000 dilution in PBST 10% BSA and incubated in a humified chamber for 60 min. The slide was rinsed again in PBST 1% BSA and deionized water for 3 min and let to air dry and then scanned on a Innopsys InnoScan 900 (Innopsys, Carbone) with the excitation 635 nm. The intensity of each spot on the microarray was analyzed and exported using the included Mapix software.

**Surface plasmon resonance interaction between glycosaminoglycans and growth factors.** From the microarray screening, 6 synthetic glycosaminoglycans were chosen for further characterization using surface plasmon resonance on the MASS-1 (Bruker, USA, Massachusetts, Billerica). The glycosaminoglycans that were chosen can be seen in Table 1, where each glycosaminoglycan has a starting sugar with an azide. those that showed high binding osaccharides, degree of sulfation, and number of carboxyl groups were counted for each glycosaminoglycan oligosaccharide.

1 μg of synthetic glycosaminoglycans was incubated with Click-iT biotin sDIBO alkyne (C20023, Thermo Fisher) at a ratio of 1:10 (molarity) at room temperature

**Table 1 Characteristics of the GAGs tested in the SPR experiment.**

| GAG | Molecular weight (Da) | Number of monosaccharides | Degree of sulfation | Number of carboxyl groups |
|---|---|---|---|---|
| GAG nr 3 | 1562.36 | 7 | 0 | 4 |
| GAG nr 10 | 2038.69 | 7 | 7 | 4 |
| GAG nr 19 | 3632.94 | 12 | 17 | 6 |
| GAG nr 26 | 2076.71 | 7 | 8 | 4 |
| GAG nr 34 | 1820.53 | 6 | 7 | 3 |
| GAG nr 43 | 2253.88 | 9 | 6 | 5 |

**Table 2 Growth factors tested and the molecular weight according to the manufacturer, the theoretical pI, the highest concentration tested for the SPR experiment as well as the lowest concentration tested.**

| Proteins tested | Molecular weight from the manufacturer (kDa) | Theoretical pI | Highest concentration tested (nM) | Lowest concentration tested (nM) |
|---|---|---|---|---|
| Beta-NGF, 256-GF-100/CF, RnD systems | 13 | 9 | 841.2 | 10.4 |
| BMP2, 355-BM-010/CF, RnD systems | 30-32 | 8.21 | 169.8 | 2.1 |
| BMP4, 314-BP-010/CF, RnD systems | 37–41 | 7.6 | 67.5 | 0.8 |
| CCL2, 279-MC-010/CF, RnD systems | 8.7 | 9.39 | 558.6 | 6.9 |
| CCL3, 270-LD-010/CF, RnD systems | 12 | 4.77 | 455.6 | 5.6 |
| CCL4, 271-BME-010/CF, RnD systems | 7.3–7.8 | 4.77 | 753.1 | 9.3 |
| CCL23, 371-MP-025/CF, RnD systems | 10.5 | 9.17 | 887.1 | 11 |
| CCN2, 9190-CC-050, RnD systems | 35–45 | 8.31 | 708.8 | 8.8 |
| CCN3, 1640-NV-050, RnD systems | 55 | 7.92 | 491.7 | 6.1 |
| CXCL2, 276-GB-010/CF, RnD systems | 8 | 9.75 | 607.5 | 7.5 |
| CXCL4, 795-P4-025/CF, RnD systems | 7.8 | 8.8 | 810 | 10 |
| CXCL7, 393-NP-010/CF, RnD systems | 7.6 | 8.79 | 729 | 9 |
| CXCL10, 266-IP-010/CF, RnD systems | 9 | 10.2 | 626.4 | 7.7 |
| CXCL12, 350-NS-010/CF, RnD systems | 7 | 9.9 | 462.9 | 5.7 |
| EGF, 236-EG-200, RnD systems | 6 | 4.78 | 810 | 10 |
| FGF1, AFL232-025, RnD systems | 16 | 7.88 | 784.7 | 9.7 |
| FGF2, ab9596, Abcam | 17 | 9.58 | 324 | 4 |
| FGF10, 345-FG-025/CF, RnD systems | 19–22 | 9.88 | 693.4 | 8.6 |
| HGF, 294-HG-025/CF, RnD systems | 60–70 | 8.23 | 124.1 | 1.5 |
| GLP1, 1851, Tocris | 4.17 | 5.05 | 777 | 9.6 |
| G-CSF, 214-CS-005/CF, RnD systems | 18.5 | 5.43 | 124.8 | 1.5 |
| IGF1, 291-G1-200, RnD systems | 7.6 | 7.76 | 810 | 10 |
| IL-6, 7270-IL-025/CF, RnD systems | 22 | 6.21 | 543.7 | 6.7 |
| IL11, 218-IL-025/CF, RnD systems | 23 | 11.16 | 602.2 | 7.4 |
| KGF, 251-KG-010/CF, RnD systems | 20 | 9.25 | 192.4 | 2.4 |
| NRG1, 5898-NR-050, RnD systems | 40–43 | 9.13 | 484 | 6 |
| PDGF-AA, 221-AA-010, RnD systems | 33 | 9.57 | 142.4 | 1.8 |
| PTN, 252-PL-050, RnD systems | 18 | 9.64 | 688.5 | 8.5 |
| SCF/c-kit ligand, 7466-SC-010/CF, RnD systems | 24–30 | 5.08 | 138.8 | 1.7 |
| SHH, 8908-SH-005/CF, RnD systems | 20 | 8.92 | 121.5 | 1.5 |
| TGF beta 1, 7754-BH-025/CF, RnD systems | 22 | 8.59 | 471.3 | 5.8 |
| TGF- beta 2, 302-B2-010/CF, RnD systems | 24 | 7.69 | 214.3 | 2.6 |
| TGF beta 3,8420-B3-025/CF, RnD systems | 18–22 | 6.1 | 526.5 | 6.5 |
| VEGF 165aa, 293-VE-010/CF, RnD systems | 39–42 | 9.01 | 121.5 | 1.5 |

overnight. The glycosaminoglycans-biotin complex was purified using HPLC (Dionex ultimate 3000, Thermo Fisher), column 3 μm C4 100 × 4.6 mm (ACE-113-1046, ACE). The immobilization was performed on a 30 nm streptavidin derivatized linear polycarboxylate hydrogel with medium charge density (SPSM SAHC30M, Xantec,), firstly conditioned with 50 mM NaOH (367176, Sigma-Aldrich) and 1 M NaCl (S7653, Sigma-Aldrich), followed by running buffer until the baseline was stable, then immobilized with 0.1 μg/ml glycosaminoglycans in PBS with 0.01% Tween 20, run for 1 min. This was repeated until the total RU value was between 10 and 25. For the runs to test protein-glycosaminoglycans interactions, the protein was dissolved in running buffer, PBS with 0.01% Tween 20, and diluted in incremental steps of 3×; starting with the lowest concentration with 2 min association time, and 10 min dissociation time with 2 min of regeneration, 50 mM NaOH and 1 M NaCl, and then repeated for all concentrations. In Table 2 we can see key characteristics for each growth factor tested as well as the highest and lower concentration tested.

Analyzing and curve fitting was done in Sierra Analyzer 3.1.36 (Bruker), with the reference spot as well as blank subtracted from each graph. This gave the $k_{on}$ and $k_{off}$ values, which the $K_d$ was calculated from, with the formula: $K_d = \frac{k_{off}}{k_{on}}$.

**ELR bioproduction and functionalization.** The design of the ELRs used in this work, termed HE5 and HRGD6, has been described elsewhere[25,52], and Table 3 includes their sequences and molecular weights. Each one comprises different bioactive domains. HE5 contains matrix metalloproteinase (MMP)-sensitive domains that allow biodegradation mainly by MMP-2, MMP-9, and MMP-13[25]. On the other hand, HRGD6 includes Arg-Gly-Asp (RGD) for cell attachment and

interaction, improving the biocompatibility of the scaffolds made with it, like the macroporous cryogels[52].

Both ELR genes were cloned into a pET-25b(+) plasmid (Novagen, Merck) for expression in *Escherichia coli* (BLR(DE3) strain, Novagen, Merck) in a 15-L bioreactor (Applikon Biotechnology)[53]. The ELRs were subsequently purified through consecutive heating and cooling steps, dialyzed against ultra-pure water and filtered through a 0.22 μm filters (Nalgene) prior to freeze-drying.

The chemical modification of the ELRs was performed following well established methods, as described by Gonzalez de Torre et al.[54]. The aim was to enable a strain-promoted azide alkyne cycloaddition (SPAAC) 'click chemistry' reaction[55] in order to achieve the cross-linking of two different ELRs to form a network that gives a stable scaffold (hydrogel or cryogel). To this end, azide ($N_3$) groups were introduced through the modification of the free ε-amine of the lysine residues present in the HRGD6 ELR, giving HRGD6-$N_3$. Similarly, cyclooctyne groups were conjugated to the HE5, to render HE5-C. Both modified ELRs were used for the SPAAC reaction and hydrogel/cryogel formation. Table 4 shows the modification degree obtained for the different ELRs.

Both the recombinantly bioproduced and the modified ELR batches were characterized following standard protocols through SDS-PAGE, H[1]-NMR, mass spectrometry and HPLC to evaluate their purity, composition, and degree of chemical modification.

**Cryogel and hydrogel production.** The ELRs were[27] dissolved in 10% (v/v) Ethanol (1015, Solveco) in PBS at a concentration of 64.28 mg/ml for the HE5-C and 35.71 mg/ml for the HRGD6-N3 and left at 4 °C overnight. MilliQ water (Millipore Reference, Merck) was sprayed using a spray bottle on top of liquid

**Table 3 Abbreviated amino acid sequence and molecular weight of the HE5 and HRGD6 ELRs were used in this work.**

| ELR | Abbreviated amino acid sequence | Mw (Da) |
|---|---|---|
| HE5 | MGSSHHHHHHHHGLVPRGSHMG-KKK-[(VPGVG)$_2$-VPGEG-(VPGVG)$_2$]$_5$-VGGGGG-PMGPSGPW-GGGG-VGGGG-QPQGLA K-GGGGG-VGGGGG-PQGIWGQ-GGGG-[(VPGVG)$_2$-VPGEG-(VPGVG)$_2$]$_5$-VGGGGG-KKK-GGGGG-[(VPGVG)$_2$-VPGEG-(VPG VG)$_2$]$_5$-VGGGGG-PMGPSGPW-GGGG-VGGGG-QPQGLAK-GGGGG-VGGGGG-PQGIWGQ-GGGG-[(VPGVG)$_2$-VPGEG-(VPG VG)$_2$]$_5$-V-KKK | 54426 |
| HRGD6 | MGSSHHHHHHSSGLVPRGSHMESLLP-([(VPGIG)$_2$-(VPGKG)-(VPGIG)$_2$]$_2$-AVTGRGDSPASS-[(VPGIG)$_2$(VPGKG)(VPGIG)$_2$]$_2$)$_6$-V | 60650 |

The sequence corresponding to the elastin-like blocks is represented by the VPGXG regions, while the lysines-containing regions (KKK for HE5 and VPGKG for HRGD6) are the cross-linking domains. Protease-sensitive (in HE5) and cell adhesion (HRGD6) domains are the regions between glycines (G) and the RGD-containing motif, respectively.

**Table 4 Modified ELRs used in this work with the number of cross-linking lysines present in the non-modified ELRs, percentage of modification, calculated number of modified lysines, and the molecular weight of the modified ELRs.**

| ELR | Modification (Mw of the 'click' group) | Original no. cross-linking lysines | % modification | No. lysines modified | Mw of modified ELR (Da) |
|---|---|---|---|---|---|
| HE5-C | Cyclooctyne (alkyne) (291 Da) | 9 | 30–40% | 3–4 | 55300–55590 |
| HRGD6-N$_3$ | Azide (228 Da) | 24 | 50–60% | 12–14 | 63400–63850 |

nitrogen and sieved through strainers with mesh sizes 500, 200, and 100 μm (Pluristrainer, 43-50500-03, 43-50200-03, 43-50100-51, pluriSelect Life Science) to collect ice crystals of desired size. Ice crystals were added to a mold (using a syringe (1 ml syringe, 329654, BD Bioscience)). Ice crystal and dissolved ELR were left in a cryostat (HM500M, Microm) at −5 °C to equilibrate. The HE5-C and HRGD6-N3 ELRs were mixed at a volume ratio of 1:1 by pipetting using cold tips, then this mixture was added to an equal volume of ice crystals and incubated at −5 °C for 15 min to fully form the cross-linkage, and then moved to −80 °C overnight. The samples were extracted from the mold and the edges were removed while being kept cold, and then samples were freeze dried (Freezone 4.5 Plus, Labconco) overnight. Hydrogels were made by dissolving each ELR in PBS at the same concentrations as for cryogels and mixed at a 1:1 volume ratio using cold tips. The mixture was pipetted into the desired molds and incubated at 4 °C for 15 min followed by 10 min at 37 °C to complete hydrogel formation.

**Scanning electron microscopy.** The hydrogels were washed 2 × 5 min in 0.1 M Sorensen´s buffer (pH 7.4) (S/3760/60 Fisher Scientific and S/4520/60, Fisher Scientific) and fixed in approximately 10 times the sample volume of "SEM fix" (0.1 M Sorensen´s phosphate buffer pH 7.4, 2% formaldehyde (P001, TAAB) and 2% glutaraldehyde (18427, Ted Pella Inc)) at room temperature for 20 min, followed by washing 2 × 5 min in 0.1 M Sorensen´s buffer pH 7.4. Samples were then dehydrated in a graded series of ethanol (50%, 70%, 80%, 90%, and twice in 100%) and subsequently critical point dried before being mounted and sputter coated with 5–10 nm Pt/Pd (80:20) (MN70-PP5708, Micro to Nano) on a turbomolecular pump coater (Q150T ES, Quorum). As cryogels were already freeze dried, they were directly mounted and coated with 5–10 nm Pt/Pd (80:20) and all samples were examined in a JSM-7800F FEG-SEM (Jeol).

**Mechanical evaluation of ELR cryogels and hydrogels.** Uniaxial compression tests were performed with an ElectroForce 5500 Test Instrument coupled to a 250 g load cell (TA Instruments). ELR cryogels with pore size 100–200 μm, 200–500 μm and 500–1000 μm (disk-shaped, 13 mm diameter × 2.5 mm height in the wet state) were formed as aforementioned and left to hydrate in PBS for 48 h at 4 °C (below the ELRs transition temperature to increase hydration efficiency). In the case of the ELR hydrogels, they were formed as described above in a Teflon mold to give disk-shaped hydrogels of 11.5 mm diameter and 1.41 mm of height in the wet state, on average. Then, each cryogel was placed on the bottom of the compression plate and the top was lowered until it touched the surface of the scaffold, which gave an increase in force sensed by the instrument, and this was set as zero force. At this point, a ramp measurement was done at 1 mm/min up to the 80% of the sample thickness (2 mm or 1.13 mm for the cryogels and hydrogels, respectively). A stress–strain curve was recorded in both cases, with the strain defined as $\varepsilon = \frac{\Delta l}{l_0}$, where $\Delta l$ is the length of compression 2 and 1.13 mm for the cryogel and hydrogel, respectively) and $l_0$ is the initial sample length or height (2.5 and 1.41 mm for the cryogels and hydrogels, respectively), and stress defined as $\sigma = \frac{F}{A_0}$, where $F$ is the force applied during compression, and $A_0$ is the area of the sample to which the force is applied (132.7 and 103.9 mm$^2$ for the cryogels and hydrogels, respectively). Stress–strain curves were drawn for the different cryogel groups and hydrogel samples with the experimental data obtained, and the tangential Young's (elastic) modulus was calculated as the slope of the stress-strain curve between 15 and 40% strain for the cryogels and 10–20% for the hydrogels (linear regions).

Moreover, oscillatory shear stress measurements were performed on the different ELR cryogels and hydrogels with the use of a controlled stress rheometer (AR2000ex, TA Instruments, New Castle, Delaware, USA) equipped with a 12 mm parallel platen and a Peltier plate temperature control that was set to 37 °C before the measurements. The gap was kept constant at 2.5 mm, and the normal force at this point was observed to be positive (0.17–0.24 N). Then, a strain sweep from 0.01 to 10% was performed at 1 Hz frequency, followed by a frequency sweep from 0.05 to 50 Hz at 1% strain. Subsequently, a time sweep measurement was performed for 3 min at 1% strain and 1 Hz (linear viscoelastic region, Supplementary Fig. 12) to obtain the storage ($G'$) and the loss ($G''$) moduli as an indication of scaffold stiffness. Data to generate the figures can be found in Supplementary Data 13–16 Three replicates of each cryogel type, i.e., of 100–200, 200–500, and 500–1000 μm pore size, or hydrogels were measured with each method. In addition, both types of scaffolds were kept hydrated during the whole experiments, which were performed in the shortest possible time to avoid drying, including re-soaking steps between the different tests. Data were analyzed in R (version 3.6.3) and is represented by the mean ± standard deviation (SD).

**HUVEC differentiation assay.** Primary HUVEC cells were purchased (C0035C, Thermo Fisher) and cultured in Endothelial Basal Medium-2 (EBM-2, CC-3156, Lonza) and supplemented with bullet kit (CC-4176) to make Endothelial Growth Medium – 2 (EGM-2, CC-3162). When passaging cells, TrypLE Express (12604013, Thermo Fisher) was used. ELR was dissolved in PBS at a concentration of 64.28 mg/ml for ELR with cyclooctyne and 35.71 mg/ml for ELR with azide and left at 4 °C overnight. Glycosaminoglycan nr 19 was added at a concentration of 2 μg/ml (for a final concentration of 1 μg/ml of ELR mixture) of ELR with cyclooctyne and left at 4 °C for 1 h. The same procedure was performed for FGF2 added at a concentration of 200 ng/ml (final concentration 100 ng/ml) and left at 4 °C for 1 h. Then the gels were mixed with equal parts ELR azide and cyclooctyne, quickly mixed by pipetting using cold pipets and 10 μl mixture was added to an angiogenesis slide (81507, Ibidi), incubated in at 4 °C for 15 min followed by 10 min at 37 °C. Primary HUVEC cells were then added at concentration of 10$^5$ per cm$^2$ in 50 μl of media. For the positive control, supplemented EBM-2 media was added, while for all the other groups non-supplemented EBM-2 was used.

**Subcutaneous evaluation of cryogels with fluorochrome in balb/cJRj mice.** The experimental procedures have been evaluated and approved by the Malmö and Lund animal experiment ethics committee (5.8.18 13902/2018). The cryogels were prepared the same way as described above with the exception that 3 μl of 10 mg/ml IRDye (680RD DBCO Infrared Dye, Li-Cor) was added to 75 μl of ELR with azide and incubated at 4 °C overnight. A small incision (0.5 cm) was made on the dorsal side below the thoracic region of female BALB/cJRj mice. A subcutaneous pocket was generated (approximated 0.5 × 1 cm) using a blunt instrument. The cryogel was implanted into the pocket and the wound was closed using a suture. The mice were housed, 5 per cage, with 12 h light-dark cycles and all animals had free access to food, water and cage enrichment. The mice were imaged in Pearl Trilogy (Li-Cor) at Excitation 685 nm and Emission 720 nm, at the timepoint directly after implantation, and after 1, 4, 6, 7, 10, and 14 days. The mice were examined daily, and the body weight was measured weekly and at termination. At this point, the mice were anesthetized and euthanized, and the implants were dissected out together with surrounding tissue and fixed in 4% PFA in PBS.

**Subcutaneous evaluation of cryogels with glycosaminoglycans and growth factor in BALB/cJRj mice**. The experimental procedures have been evaluated and approved by the Malmö and Lund animal experiment ethics committee (5.8.18 02361/2019). The cryogels were prepared the same way as described in the cryogel production with the exception that the cryogels with the glycosaminoglycan GAG nr 19 was added at a concentration of 25 µg/ml (for a final concentration of 12.5 µg/ml of ELR mixture) to a solution containing the ELR-cyclooctyne and left in the fridge overnight. After freeze drying, the cryogels were autoclaved and resuspended in 200 µl of either pure PBS, 200 ng/ml VEGF, 1 µg/ml of FGF2 or both 200 ng/ml VEGF and 1 µg/ml FGF2. Cryogels were left at 4 °C for two days and washed with PBS before implantation. The testing groups were control, GAG, VEGF, VEGF GAG, FGF2, FGF2 GAG, VEGF FGF2 and VEGF FGF2 GAG at time points 2, 4 and 8 weeks with 5 mice in each group and time point. Female BALB/cJRj mice were anaesthetized with isoflurane and received a subcutaneous injection of Marcain (4 mg/kg) close to the incision site. A small incision (0.5 cm) was made on the dorsal side below the thoracic region of the mice. A subcutaneous pocket was generated (approximated 0.5 × 1 cm) using a blunt instrument. The cryogel was implanted into the pocket and the wound was closed using sutures. Only one cryogel were implanted per mouse. The mice were housed 5 per cage with 12 h light dark cycles and all animals had free access to food, tap water and cage enrichment. The mice were studied daily, and weekly regarding body weight measurement and at termination. At termination, the mice were anesthetized with isoflurane and blood was collected through heart puncture and transferred to Li-Heparin coated vials. After centrifugation, the plasma was transferred to Eppendorf tubes and stored at −80 °C. The implant was dissected out together with surrounding tissue and was fixed in 4% PFA in PBS.

**Preparation for histology**. All samples were dehydrated, cleared and infiltrated and embedded in paraffin automatically in a TISSUE-TEK V.I.P. (Miles Scientific). 4 µm paraffin sections were prepared with a microtome and dried in oven at 37 °C overnight. For hematoxylin & eosin staining the sections were deparaffinated and stained with Mayer's hematoxylin for 6 min. (05-06002/L, Bio-Optica), washed in running tap water for 10 min, then stained with eosin for 3 min (05-10003/L, Bio-Optica) and washed in distilled water, dehydrated, and mounted with coverslips. For CD31 staining, the samples were deparaffinated and antigen retrieval pretreatment of 20 min boiling at 97 °C in pH 9.0 (S2367, Dako) and 20 min of cooling at room temperature were performed. The slides were washed in PBS for 5 min and pre-blocked in 10% normal mouse serum (015-000-120, Jackson Immunoresearch) for 20 min, and the primary antibody (CD31, ab28364, Dilution: 1/50, Rabbit anti Mouse, Abcam) was added for 60 min with 5% normal mouse serum. Subsequently, it was washed 3 times in PBS and the secondary antibody BrightVision rabbit/HRP (DPVR110HRP, No dilution (Ready to use), Immunologic), was added for 30 min. Washing was performed 3 times in TRIS buffer and 3,3′-Diaminobenzidine (DAB) was added for 5 min. The samples were then rinsed, counterstained in hematoxylin for 20 s, dehydrated and mounted. For macrophage staining, samples were deparaffinated and antigen retrieval pretreatment of citric acid pH 6.1 for 30 min at 97 °C (K8005, Dako) and cooled down to 65 °C. The slides were washed in PBS for 5 min (three times) and the primary antibody (for M1 staining iNOS, ab3523, Dilution: 1/100, Rabbit anti Mouse, Abcam and F4/80, ab16911, Dilution: 1/25, Rat anti Mouse, Abcam same for M2, but replacing iNOS with CD206, ab64693, Dilution: 1/200, Rabbit anti Mouse, Abcam) was diluted in 2% BSA and added for 90 min, washed in PBS for 5 min (three times) and the secondary antibodies (IgG Alexa 647 plus, A32733, Dilution: 1/500, Goat anti Rabbit, Thermo Fisher and IgG Alexa 555 plus, A48263, Dilution: 1/500, Goat anti Rat, Thermo Fisher and DAPI, MBD0015, Dilution: 1/1000, Sigma Aldrich) were added and incubated for 45 min in the dark, washed for 5 min (three times) and mounted with Dako fluorescent mounting medium (S3023, Dako).

**Evaluation of histology**. All brightfield slides were scanned on VS120 (Olympus Corporation, Tokyo, Japan) at 40×. For brightfield, CD31 analysis Visiopharm (VIS 20190808) was used and for each slide the cryogel was marked in the software and every cell nucleus was counted as well as each DAB positive cell. The number of positive DAB cells were then divided by the total area of the cryogel analyzed, to remove background the same procedure was performed on the control staining without primary antibody and subtracted from the sample. Analysis of giant multinucleated cells were made by scoring either present (1) or absent (0), it also overlapped with the ELR breaking down and visually seeing pieces of ELR. For the florescent macrophage stainings, all the slides were scanned on VS120 at 20x and analyzed using QuPath 0.2.3[56], manually excluding large blood clots from the analysis, otherwise the whole slide was examined. Then every nuclei (DAPI) was detected and all double positive cells were counted and normalized by dividing by total number of cells for the M1 and M2 slide. The normalized M2 macrophage count was then divided by the sum of the normalized number of M1 and M2. (M2/(M1 + M2)). All bar graphs were plotted in GraphPad Prism 9 (GraphPad Software).

**Multiplex ELISA of plasma**. Mouse Magnetic Luminex Assay was bought from Bio-Techne and plasma samples were run in duplicates or singlets depending on available material. All samples were centrifuged at 16,000 × g for 4 min before dilution. All samples were diluted 1:2 according to manufacturer's recommendation, 50 µl of diluted were added to each well together with 50 µl of microparticle cocktail and incubated on a shaker at 800 rpm, at 4 °C, overnight. The plates were then attached to a magnet and washed three times. 50 µl of Antibody cocktail was added and incubated at room temperature on the shaker for 1 h. The washing was repeated. Streptavidin-PE was added at 50 µl per well and incubated at room temperature for 30 min on a shaker. The washing was repeated again and reconstituted in 100 µl of wash buffer and analyzed on a Luminex MAGPIX (Luminex).

**Statistics and reproducibility**. Each growth factor screen on the microarray was screened once. The data is shown in a box and whisker (as Tukey) of 36 binding spots for each synthetic glycosaminoglycans.

All growth factors for the SPR experiments were run at 5 different concentrations and the trueness of the fit, chi2 is available in the supplement for each growth factor.

For the mechanical testing, stress–strain curves and rheological measurements of cryogels with different pore sizes are represented as mean ± SD; $n = 3$.

For the in vivo experiment 5 mice were used in each group, but some samples were not found when extracted, number above each bar indicates the n number used. For the Luminex assay, plasma was collected from all animals. The data is plotted as mean with ± SD.

**Reporting summary**. Further information on research design is available in the Nature Research Reporting Summary linked to this article.

## Data availability

The datasets generated and/or analyzed during the current study are found in the Supplementary Data where all other data is available from the corresponding author upon reasonable request.

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

## Acknowledgements

Lund University Bioimaging Center (LBIC), Lund University is acknowledged for providing experimental resources. In vivo studies were caried out together with Truly Translational, Lund. We would like to thank Lisa Karlsson, Lung Biology, Lund University, for the initial illustration of experimental design (2A, 3A, and 4A) which has then been changed to reflect each experiment. Founding was generosity provided by: Swedish Heart Lung foundation grant 20210086 (G.W.T.). Swedish Research Counsel grant 01190, 01375 (G.W.T.). Swedish Foundation for Strategic research grant SBE13-0130 (G.W.T.). Lund University Medical Faculty (G.W.T.). ALF project 0097 (G.W.T.). Ewy o Gunnar Sandberg Foundation (G.W.T.). Åke and Inger Bergkvist foundation (L.E.R.). The Royal Physiographic Society of Lund grant 40451 (Z.S. and E.T.). Crafoord Foundation (LER). LMK Postdoc 2019 (E.T.). Sten K Johnsons stiftelse (SH). Spanish Government (PID2019-110709RB-100, RED2018-102417-T) (J.C.R.C.). Junta de Castilla y León (VA317P18, Infrared2018-UVA06) (J.C.R.C.). Interreg V España Portugal POC-TEP (0624_2IQBIONEURO_6_E) (JCRC). Centro en Red de Medicina Regenerativa y Terapia Celular de Castilla y León (J.C.R.C.). Fundación Ramón Areces (grant submission number BEVP33S12276) (AIF).

## Author contributions

Z.S.: Conceptualization, methodology, investigation writing - original draft, visualization, and formal analysis. A.I.-F.: Investigation, methodology, and writing - original draft. S.H.: Investigation, methodology, funding acquisition, and writing - review & editing. J.C.R.-C.: Resources and writing - review & editing. J.L.: Resources and writing - review & editing. L.Y.: Resources and writing - review & editing. E.T.: Supervision, investigation, methodology, conceptualization, writing - original draft, and funding acquisition. L.E.R.: Supervision, conceptualization, methodology, writing - original draft, and funding acquisition G.W.-T.: Resources, supervision, conceptualization, funding acquisition, and writing - original draft.

## Funding

## Competing interests
The authors declare the following competing interests: J.L. is founder of Glycan Therapeutics LLC. The remaining authors declare no competing interests.
