## [Peer Review File · Communications Biology]

Reviewers' comments:

Reviewer #1 (Remarks to the Author):

Söderlund et al., have created an array of synthetic GAGs and shown the binding of a variety of proteins, with particular focus on VEGF and FGF2. By using ELR functionalised with promising GAG candidates, they have shown functionalisation of the cryogels with these two proteins and shown vastly improved blood vessel formation in vivo. This work is of a very high standard, it is well-written and well-presented. It will clearly be of interest to a broad range of researchers in the TERM field. I therefore recommend it for publication with a few minor changes.

1) There are some areas of the methodology that need to be improved for better reproducibility of the work. The authors should check carefully that ALL reagents and equipment used is described in full with manufacturer's identity etc. One example of this is in the section on cryogel formation. How was the water sprayed? Which device was used? Which filter manufacturer was used for this work etc. These details should be included before publication – making it possible for someone else to reproduce this.

2) This is a recommendation rather than a requirement, but I strongly recommend that the authors perform the compression analysis and the rheology for the hydrogel control sample. They will probably find that this is much stiffer than the cryogel counterparts, but it is important to include as a control, not so much for full characterisation of the materials, but to act as a comparison for the cryogel properties. It was included in the SEM so could be included in the mechanical testing.

3) I know the focus of this work has been on the GAG synthesis and characterisation (which is very impressive), however, since the proof of concept and the very interesting in vivo work was carried out using cryogel scaffolds, I think it would be in improvement to the manuscript if a small section was included in the introduction about cryogel systems that have been used for protein delivery.

This could include (but of course not limited to) two interesting papers:

Koshy et al., Injectable nanocomposite cryogels for versatile protein drug delivery, 2018 Acta Biomaterialia, Volume 65, January 2018, Pages 36-43

And

Newland et al., Injectable Glycosaminoglycan-Based Cryogels from Well-Defined Microscale Templates for Local Growth Factor Delivery, ACS Chem. Neurosci. 2021, 12, 7,

4) Lastly, we think that the authors point in the discussion is very true and important, so this line should be backed up with a reference: "replenish the site with oxygen and nutrients and is often the first hurdle to overcome to have a successful graft". The authors recommend a recent review by Rizzo et al., which describes oxygen-glucose deprivation hampering graft survival: Rizzo et al., Oxygen-glucose deprivation in neurons: implications for cell transplantation therapies, Progress in Neurobiology, Volume 205, October 2021, 102126

Reviewer #2 (Remarks to the Author):

Comments:

In the paper titled "Synthetic glycosaminoglycans and macroporous elastin-like recombinamers as a modular platform for tissue regeneration" the authors have synthesized glycosaminoglycans (GAGs) and showed their binding with a variety of growth factors at different K_d , K_{off} and K_{on} . Further, a synthetic GAG (GAG nr19) was selected based on its binding to various growth factors. A microporous structured scaffold containing GAG nr 19 was then fabricated for controlled release of growth factors from scaffolds for blood vessel formation.

Various synthetic GAGs (such as sulfated alginate and sulfated carboxymethyl cellulose) have been utilised to overcome the limitations of low retention of growth factors in scaffolds. However, the present study screened wide range of synthetic glycosaminoglycans that can be used for growth factor binding. The study has implications in the field of tissue engineering and regenerative medicine. It provides an array of synthetic GAGs and a method to select a suitable combination of GAG and growth factor to control the release of growth factors from scaffolds for a desired application.

The manuscript needs the following major improvements:

1. Line 116-117: It is not clear what does a unique binding of growth factors to GAGs mean? The authors must elaborate on the rationale for selection of GAG nr 10, 26, 34 and 43 for SPR studies.
2. The authors used VEGF and FGF in the functional studies of the manuscript, however, they screened three groups of GAGs from microarray using VEGF and HGF. Was there a specific reason for choosing HGF and not FGF for screening GAGs.
3. In the HUVEC differentiation study, why was no differentiation observed in the group containing free FGF2? The result from the experiment suggests that the differentiation is due to the presence of synthetic GAG and not due to the growth factor (FGF2). This seems to be contrary to what has been observed subcutaneously in in vivo experiment.
4. In results, line 237, the authors have discussed about cryogel size and its presence in tissues other than at implantation site, however, the authors have not shown results for day 1, 4, 6 and 10.
5. Line 243-244: The authors need to verify the units. It does not match with the units in figure 4C.
6. The supplementary figures have not been discussed in the results section.
7. The authors must discuss all the results in section 2.4 of the manuscript in detail. The authors have selectively discussed few results and chose not to discuss others in section 2.4.
8. It would be convenient for readers if authors can include results for IL-6, VEGF, FGF2, MMP8 and TIMP-1 as column graphs in supplementary information, instead of providing enormous data in a table format.
9. In supplementary figure 2, 3 and 4, the components of 'naïve group' is not clear.
10. The language used in the paper needs to be significantly improved to increase clarity. In addition, the entire manuscript needs to be thoroughly reviewed for spellings and typographical errors.

Reviewer #3 (Remarks to the Author):

Manuscript Number: COMMSBIO 22 1236 T

The authors investigated the potential of chemoenzymatically synthesized glycosaminoglycans (GAGs) for controlled binding and release of growth factors. After screening a library of 52 GAGs, selected synthetic GAGs were used to functionalize nanoporous or macroporous hydrogels based on elastin-like recombinamers (ELRs). These materials were loaded with pro-angiogenic growth factors to direct cellular response in a controlled manner. As a proof-of-concept (a) in vitro the tubular morphogenesis of endothelial cells seeded on hydrogels and (b) in vivo the tissue response upon subcutaneous implantation of macroporous hydrogels into a mouse model were studied. The idea and general concept of incorporating synthetic GAGs into hydrogels to modulate growth factor concentration is interesting and would overcome many problems of hydrogels based on naturally derived GAGs that are highly heterogeneous [doi.org/10.1021/acs.accounts.9b00420].

However, the authors should nevertheless mention in the manuscript that naturally derived unmodified and modified GAGs have already been successfully used for many years in hydrogels for such purposes - relevant papers should be cited (please also compare Specific comments). It has also been shown for naturally derived GAGs (often heparin) that by varying the degree of sulfation of the GAG and/or the GAG concentration in the hydrogel, uptake and release of signalling proteins can be controlled (e.g., doi.org/10.1016/j.biomaterials.2018.07.056, doi: 10.1002/ADFM.202000068, doi: 10.1039/C9FD00016J). Macroporous GAG-based hydrogels were also frequently used for sustained protein delivery (e.g., doi: 10.1016/J.BIOMATERIALS.2021.121170), for instance to induce vascularization like in the paper under consideration (doi: 10.1016/j.actbio.2014.11.002, doi: 10.1002/mabi.202100234 or doi: 10.1021/acs.biomac.8b00331). This "State of the Art" should be added to the manuscript. Another major criticism is that the authors do not show whether the tunability of binding and release of growth factors by the synthetic GAGs that they found for surface-bound GAGs is maintained when the GAGs are incorporated into hydrogel networks and whether similar trends are found here as for the surface-bound GAGs. As only a single GAG was selected to produce the hydrogel materials used for the in vitro and in vivo experiments, the advantages and a convincing evidence of the modularity of a platform based on different synthetic GAG-based hydrogels are not really shown experimentally. Regarding the ELR component of the hydrogels the authors mention in the introduction that "Due to the recombinant expression, several modifications can be made, such as introducing a RGD motif for attachment, a matrix metalloproteinase domain for controlled degradation or functionalization of lysine groups for further modification." (page 5, lines 98-100), but later do not further explain the properties of the specific ELR that they used in their study (see below). Here, too, it would have made sense to compare at least two different ELRs to really demonstrate the modularity of the material system.

The Methods part should please be improved.

Some calculations and interpretations are not comprehensible to me or wrong in my opinion (details see below).

Specific comments:

1. The title could have been chosen more appropriately to emphasize that this study also reports about biomaterial development. Moreover, it is not the elastin-like recombinamers that are macroporous, but the hydrogels made from them.

2. The selection of the abbreviations listed and explained seems arbitrary. Why, for example, is only the abbreviation of some growth factors explained?

3. The wording of the abstract could be more precise. In particular, I would also mention that K_d values were determined. I suggest naming the technique used to introduce the macropores not as cryogelation; it is rather a combination of porogen leaching and cryogelation. In the abstract the authors do not mention that they also tested VEGF loaded macroporous hydrogels for the in vivo studies.

4. The introduction could be more conclusive and include more references to the current "State of the Art". The authors mention only few examples of how in general sustained release from biomaterials can be achieved. More strategies reported in original articles and review articles could be added here. For instance, the review article doi: 10.1039/C3TB20853B summarizes "Chemical strategies for the presentation and delivery of growth factors". The strategy pursued in nature of adjusting growth factor gradients via GAGs, which the authors describe as very advantageous, has also been used in the design of biomaterials to modulate growth factor concentrations and is in principle not new (doi:10.3389/fcell.2021.760532). The authors should mention this and cite some relevant previous studies, e.g. doi: 10.1016/j.actbio.2014.11.002, doi: 10.1002/mabi.202100234, doi: 10.1002/ADFM.202000068, doi: 10.1021/acs.biomac.8b00331 or doi: 10.1016/J.BIOMATERIALS.2021.121170, to name only some examples of heparin-based hydrogels.

From the introduction in its present form the exact intention of the study is not clear. It is argued somewhat arbitrarily with various aspects of ECM, which are apparently to be imitated by the materials presented here. Why both nanoporous and macroporous systems were investigated is not plausible for me. In part, the impression is given that macroporous hydrogels are generally better suited for tissue regeneration and that their too fast release of growth factors is a problem

that should be solved using principles of nature, i.e. by creating GAG-based systems. A specific comparison of the introduced materials with and a differentiation from already established GAG-based hydrogel systems is missing.

Some explanations are not coherent from my point of view: Page 4, line 70/71: „... by having growth factors incorporated into scaffolds such as hydrogels, where the release is limited due to the low area to volume ratio.“

5. Characterization of GAG-growth factor interactions:

a. Although precise information on the sequences of the various synthetic GAGs can be found in the SI, it would be helpful for the reader to mention some information on the GAGs that could play a role in growth factor binding directly in the manuscript, e.g. as a table summarizing the degree of sulfation, the molar mass, the number of negatively charged groups (e.g. also carboxyl groups that could mediate the binding of growth factors to non-sulfated GAGs, such as GAG no. 3, see p. 6, line 130/13). So far, only the numbers assigned to the GAGs are given here, or the sequences of the selected GAGs are shown very small in the axis in Figure 1. In addition, the specification of important properties of the growth factors used, such as molar mass, IEP, dimensions, would help to better interpret/understand the interaction with the GAGs. The analysis should therefore be based more on physicochemical properties of the components and be comprehensible to the reader through a systematic presentation. In particular, a comparison with the only natural GAG heparin (no. 1) would perhaps also be interesting here (or at least a hint that heparin is hidden behind no. 1), whereby knowledge of the molar mass (distribution) used would also be interesting here and a comparison with literature data (e.g. with doi:10.3390/molecules24183360).

b. On page 6, lines 114-117, the authors state "The growth factors added to the microarray revealed three groups of glycosaminoglycans: those that showed high binding to all growth factors, those that showed low/no binding and finally those that showed unique binding depending on the growth factor. For group 2, information is given in the legend to Figure 1. For the other two groups, corresponding information should be please given as well. One possibility could be to include it in the additional table on the GAG characteristics.

c. In addition, there is a certain discrepancy between the number of GAGs examined in the text (52), in the table (nr. 1-53=54) and in Figure 1 (nr. 0 - 53=54). A clear allocation of the GAGs in the figures is therefore not possible.

d. Is the use of equal mass concentrations of the different growth factors useful in microarray screening, as is the use of equal mass concentrations of the antibodies? Would not an equal molar concentration of substances make more sense? What protein concentrations were used for SPR? I could not find any information on this in the manuscript.

e. The formula given on page 25 is not correct in my opinion. It should correctly read $K_d = K_{off}/K_{on}$. Furthermore, the authors should definitely check all the values for K_d given in the SI (12692_0_data_set_98338_r3frs0.xlsx). I calculated different values except for the two growth factors CXCL12 and CXCL4. This would also change the presentation in the Results or Discussion Part in some cases. For example, for GAG nr. 19 I do not calculate the same K_d for VEGF (410 nM) and (FGF-2 (413 nM), as the authors state and interpret on page 7, line 137 and page 20/21, lines 397, but a 3 times higher K_d for VEGF (136 nM) than for FGF-2 (51 nM) underpinning the higher effect for FGF-2 found in the cell experiment. It is questionable whether the statement that "focus should be on release rate and not total binding when looking to sustain the efficacy of growth factors when using synthetic glycosaminoglycans" can still be sufficiently substantiated.

f. Due to the incorrect calculation of the K_d values, the factor 2000 on page 18, line 340 ff, is incorrect in my opinion, too. The correct value would be about 1000.

g. The claim made on page 6, lines 124-127 "The synthetic glycosaminoglycan that had the longest carbohydrate chain as well as the highest degree of sulfation, GAG nr 19 (highlighted in orange in Fig. 1E), showed unique binding to transforming growth factor beta 1 (TGF beta 1), keratinocyte growth factor (KGF) and connective tissue growth factor (CCN2)". I cannot understand this on the basis of Figure 1. Is it more likely that the following is meant: "The synthetic glycosaminoglycan with the longest carbohydrate chain as well as the highest degree of sulphation, GAG No. 19 (highlighted in orange in Fig. 1E), was the only one to show binding to transforming growth factor beta 1 (TGF beta 1), keratinocyte growth factor (KGF) and connective tissue growth factor (CCN2)"?

6. Production of the hydro-/cryogel materials:

a. The description of hydro- and cryogel production should be optimized. In the current way it is

not understandable and not comprehensible.

b. It should be explained very briefly how the ELRs are produced, even if this work has already been published. I would recommend including a brief description of the properties (e.g., molecular weight, RGD sequences, functionalisation with cyclooctyne and azide).

c. Instead of using mass concentrations and ratios, it would be probably better to work with molar quantities in some cases.

d. It is not really clear to me which functional groups are involved in the hydrogel formation and functionalization, especially in the incorporation of growth factors and GAGs, and what the molar ratios of ELR : growth factor : GAG are. Is the growth factor added during the production of the hydrogel materials or are the materials loaded with the proteins afterwards? Apparently, this is handled differently for the hydrogels and the cryogels.

e. The cross-linking chemistry for the different systems should be please explained more clearly in a scheme. So far, single important information is given in the Results, Discussion and Methods Part and the reader need somehow to combine it.

7. Characterization of the hydro/cryogel materials:

a. So far the authors present the mechanical characterization of only the pure ELR cryogels with different pore sizes (two methods were used to determine storage modulus and Young's modulus; the values obtained are not compared or discussed further). A comparison with GAG-containing cryogels would have been useful to see if/how the GAG changes the mechanical properties of the materials.

b. Also, the pore structure is only shown for the pure ELR cryogels and not for the GAG-containing ones, which are then ultimately used in the in vivo experiment.

If it is not known whether the pore structure and the mechanical properties are the same as in the control, it cannot be ruled out that changes in the cell response also result from a changed overall or local stiffness (which is not determined at all) or pore geometry and not only from the release of the growth factor.

c. It is a nice result that macroporous hydrogels (cryogels) with different pore sizes can be produced, but the influence of the pore size on the in vivo response is not investigated (cf. also above, influence of the GAG component).

d. For the sake of completeness, the mechanical properties of the ELR and ELR-GAG hydrogels should also be determined and shown.

e. In my opinion, the most important point missing regarding the characterization of the ELR and ELR-GAG hydrogel materials to substantiate the study and make it convincing are studies on the uptake and release of the growth factors in the GAG-modified hydrogels and cryogels. The binding studies on the surface-bound GAGs by means of the screening assay and SPR can certainly only provide initial indications of the binding and release behavior of various growth factors to GAG-containing hydrogel materials, but cannot be used alone to interpret the in vitro and in vivo data shown and cell-instructive properties of the hydrogel materials in general. It is expected that the comparison of uptake and release of VEGF and FGF-2 from GAG nr. 19-containing cryogels will provide important clues to better understand the observed in vivo response.

f. Why is the combination of FGF-2 with additional VEGF only investigated in the cryogels/in vivo and not in the hydrogels/in vitro experiments?

g. The addition of VEGF showed no significant effect in the in vivo experiment. However, only 1/5 of the mass of FGF-2 was used to load the cryogels. Furthermore, the loading efficiency may be different for the two factors. However, this was not determined. The analysis of the release behavior of the two different factors from the cryogel materials could be more helpful in the interpretation of the data than the comparison of the K_d values determined on surface-bound GAGs.

h. Did the authors check whether autoclavation changes the properties of the cryogels?

8. The authors should explain why they use a hydrogel for the in vitro experiments and a macroporous cryogel for the in vivo experiments. Why is the morphogenesis of HUVECs studied in 2D and not in 3D in the hydrogel system (especially since the abstract also mentions 3D in vitro systems)?

9. The technical terms should be used correctly. Firstly, it is confusing when a capital K is used for both the thermodynamic quantity equilibrium dissociation constant (K_d) and the kinetic constants K_{on} and K_{off} . Usually, a small k is used for the kinetic velocity constants. Also, it should correctly read "equilibrium dissociation constant", "dissociation rate constant" and "association rate constant" instead of "total binding" or "equilibrium constant", "dissociation constant" and

"association constant". Accordingly, on page 6, line 118, it should also correctly read " ... on a kinetic and thermodynamic level ...".

I think the authors do not mean "naïve" but "native" and not "capsel" but "capsule (e.g. Figure SI1, I)".

10. Some statistical data are missing, for instance in Figure 1 A and C the authors do not indicate what is plotted here and what is shown by the error bars. Please add to the Figure Legend or add a separate section "Statistics", where you give this important information. Similarly, for the determination of K_d , K_{on} and K_{off} , the number of measurements and the errors are not given (neither in the manuscript nor in the tables in the SI). Was only one experiment carried out?

11. The References should be in a uniform format. Ref. 14 contains two-times the term "Author manuscript". Sometimes doi numbers are given, sometimes not.

12. The labels in the figures are in part clearly too small.

Reviewers' comments together with answers:

Reviewer #1 (Remarks to the Author):

Söderlund et al., have created an array of synthetic GAGs and shown the binding of a variety of proteins, with particular focus on VEGF and FGF2. By using ELR functionalised with promising GAG candidates, they have shown functionalisation of the cryogels with these two proteins and shown vastly improved blood vessel formation in vivo. This work is of a very high standard, it is well-written and well-presented. It will clearly be of interest to a broad range of researchers in the TERM field. I therefore recommend it for publication with a few minor changes.

1) There are some areas of the methodology that need to be improved for better reproducibility of the work. The authors should check carefully that ALL reagents and equipment used is described in full with manufacturer's identity etc. One example of this is in the section on cryogel formation. How was the water sprayed? Which device was used? Which filter manufacturer was used for this work etc. These details should be included before publication – making it possible for someone else to reproduce this.

Thank you for this valuable comment, the method section has been updated and details have been added. For example, this section:

Water was sprayed using a on top of liquid nitrogen and sieved through strainers with mesh sizes 500, 200, and 100 to collect ice crystals of desired size. Ice crystals were added to a mold together with the dissolved ELR and left in a cryostat at -5 °C to equilibrate.

Has been changed to: (Line: 624 -629)

MilliQ water (Millipore Reference, Merck) was sprayed using a spray bottle on top of liquid nitrogen and sieved through strainers with mesh sizes 500, 200, and 100 μm (Pluristrainer, 43-50500-03 ,43-50200-03, 43-50100-51, pluriSelect Life Science) to collect ice crystals of desired size. Ice crystals were added to a mold (here a syringe was used (1 ml syringe, 329654, BD Bioscience) kept at -5°C together with the dissolved ELR and left in a cryostat (HM500M, Microm) at -5 °C to equilibrate.

2) This is a recommendation rather than a requirement, but I strongly recommend that the authors perform the compression analysis and the rheology for the hydrogel control sample. They will probably find that this is much stiffer than the cryogel counterparts, but it is important to include as a control, not so much for full characterisation of the materials, but to act as a comparison for the cryogel properties. It was included in the SEM so could be included in the mechanical testing.

We thank the reviewer for the recommendation, which we have followed. The results obtained can be found as Supplementary figures in the new version of the manuscript. As

expected by the reviewer, the hydrogels were found to be stiffer than the cryogel counterparts, which is described in the results section of the manuscript as follows: (Line 255-258, and 652-687)

On the other hand, the ELR hydrogel gave a tangential E value of 4041.6 ± 325.8 Pa (Supplementary Figure 3), while the G' was found to be 633.8 ± 95.9 Pa (see Supplementary Figure 4 for the strain sweep curve). Both measurements resulted in a much higher stiffness than for all the groups of the cryogel counterparts.

The methods and discussion sections have also been updated to include the mechanical measurements of the ELR hydrogels.

3) I know the focus of this work has been on the GAG synthesis and characterization (which is very impressive), however, since the proof of concept and the very interesting in vivo work was carried out using cryogel scaffolds, I think it would be in improvement to the manuscript if a small section was included in the introduction about cryogel systems that have been used for protein delivery.

This could include (but of course not limited to) two interesting papers:

Koshy et al., Injectable nanocomposite cryogels for versatile protein drug delivery, 2018 Acta Biomaterialia, Volume 65, January 2018, Pages 36-43

And

Newland et al., Injectable Glycosaminoglycan-Based Cryogels from Well-Defined Microscale Templates for Local Growth Factor Delivery, ACS Chem. Neurosci. 2021, 12, 7,

Thank you for these interesting articles, these have been added to the introduction together with this text: (Line : 105-110)

Multiple methods exist to create macroporous structures, though one of the more common ones is cryogels, where the material forms around ice crystals. When the ice is removed one is left with a macroporous structure with interesting characteristics such as shape-memory properties that makes it injectable through a syringe.

4) Lastly, we think that the authors point in the discussion is very true and important, so this line should be backed up with a reference: "replenish the site with oxygen and nutrients and is often the first hurdle to overcome to have a successful graft". The authors recommend a recent review by Rizzo et al., which describes oxygen-glucose deprivation hampering graft survival: Rizzo et al., Oxygen-glucose deprivation in neurons: implications for cell transplantation therapies, Progress in Neurobiology, Volume 205, October 2021, 102126

Thank you for this very informative review which highlights the importance of oxygen for graft integration

Reviewer #2 (Remarks to the Author):

Comments:

In the paper titled "Synthetic glycosaminoglycans and macroporous elastin-like recombinamers as a modular platform for tissue regeneration" the authors have synthesized glycosaminoglycans (GAGs) and showed their binding with a variety of growth factors at different K_d , K_{off} and K_{on} . Further, a synthetic GAG (GAG nr19) was selected based on its binding to various growth factors. A microporous structured scaffold containing GAG nr 19 was then fabricated for controlled release of growth factors from scaffolds for blood vessel formation.

Various synthetic GAGs (such as sulfated alginate and sulfated carboxymethyl cellulose) have been utilized to overcome the limitations of low retention of growth factors in scaffolds. However, the present study screened a wide range of synthetic glycosaminoglycans that can be used for growth factor binding. The study has implications in the field of tissue engineering and regenerative medicine. It provides an array of synthetic GAGs and a method to select a suitable combination of GAG and growth factor to control the release of growth factors from scaffolds for a desired application.

The manuscript needs the following major improvements:

1. Line 116-117: It is not clear what does a unique binding of growth factors to GAGs mean? The authors must elaborate on the rationale for selection of GAG nr 10, 26, 34 and 43 for SPR studies.

From:

Fig. 1 A-C show vascular endothelial growth factor (VEGF) and hepatocyte growth factor (HGF) as an example of the results obtained with this method. The growth factors added to the microarray revealed three groups of glycosaminoglycans: those that showed high binding to all growth factors, those that showed low/no binding and finally those that showed unique binding depending on the growth factor.

To further study this on a kinetic level, we decided to evaluate a subset of the glycosaminoglycans (one low/no binding, GAG nr 3, one high binding, GAG nr 19, and four with unique binding, GAG nr 10, 26, 34 and 43) using surface plasmon resonance (SPR) (see Supplement Table 2).

To: (Line 151-164)

Fig. 1 A-C show vascular endothelial growth factor (VEGF) and hepatocyte growth factor (HGF), furthermore, BMP4, CXCL12, IL-6, FGF2, PDGF-AA, KGF and TGF beta 1 was also tested, and data can be found in Supplement Figure 1. The growth factors added to the microarray revealed three groups of glycosaminoglycans: those that showed high binding to all growth factors, those that showed low/no binding and finally those that showed binding depending on the growth factor.

To further study this on a kinetic level, we decided to evaluate a subset of the glycosaminoglycans (one with low to no binding to any growth factors, GAG nr 3, one high binding (except for TGF beta 1), GAG nr 19, and four with binding to only a subset of the

growth factors, GAG nr 10, 26, 34 and 43) using surface plasmon resonance (SPR) (see Supplement Table 2).

2. The authors used VEGF and FGF in the functional studies of the manuscript, however, they screened three groups of GAGs from microarray using VEGF and HGF. Was there a specific reason for choosing HGF and not FGF for screening GAGs.

The supplement data for microarray screening of BMP4, CXCL12, IL-6, FGF2, PDGF-AA, KGF and TGF beta 1, has been added to supplement figure 1

3. In the HUVEC differentiation study, why was no differentiation observed in the group containing free FGF2? The result from the experiment suggests that the differentiation is due to the presence of synthetic GAG and not due to the growth factor (FGF2). This seems to be contrary to what has been observed subcutaneously in in vivo experiment.

It has previously been shown that glycosaminoglycans, FGF and the FGF receptor form a complex which is needed for activation and thus there is a need for all three molecules to be present for an activation to occur.

More can be read about glycosaminoglycans and its interaction with receptors and growth factors in this great review:

“Mechanistically, HS directly interact with both FGF and their receptors (FGFR) by forming FGF-FGFR-HS ternary complex on the cell surface. Functionally, HS serves as a co-receptor to facilitate the FGF-induced FGF receptor dimerization and activation”

<https://www.sciencedirect.com/science/article/pii/S0898656818303000#bb0165>

4. In results, line 237, the authors have discussed about cryogel size and its presence in tissues other than at implantation site, however, the authors have not shown results for day 1, 4, 6 and 10.

From:

Prior to this, a pilot study was performed to assess the timespan of the experiment with a focus on degradation of ELR cryogels, to compare with the results from a previous in vivo study with ELR hydrogels 17. Cryogels were labelled with the 680RD fluorescent infrared dye, implanted subcutaneously and monitored at 1, 4, 6, 7, 10 and 14 days with no difference in size observed and no detectable cryogel elsewhere (Fig. 4B)

To: (Line 282 -288)

Cryogels were labelled with the 680RD fluorescent infrared dye at different concentrations to determine the optimal dilution. The highest brightness was observed for 2 μg dye per 50 (100 times dilution) mg cryogel (Supplement Figure 5A) and was further used for cryogels that were implanted subcutaneously. The labeled cryogels were monitored at 1, 4, 6, 7, 10 and 14 days with no difference in size and intensity observed (Supplement figure 5 B and C) and no detectable cryogel elsewhere (Fig. 4B as well as Supplement figure 5C).

Supplement figure 2 has now been added.

5. Line 243-244: The authors need to verify the units. It does not match with the units in figure 4C.

Thank you for spotting this error, μm^{-1} has been changed to μm^{-2}

6. The supplementary figures have not been discussed in the results section.

The supplement figures have been more extensively discussed in section 2.4 (The only section which had supplement figures before revision)

7. The authors must discuss all the results in section 2.4 of the manuscript in detail. The authors have selectively discussed few results and chose not to discuss others in section 2.4.

To the best of our knowledge, we have discussed all results but are happy to add additional information.

8. It would be convenient for readers if authors can include results for IL-6, VEGF, FGF2, MMP8 and TIMP-1 as column graphs in supplementary information, instead of providing enormous data in a table format.

Supplement figure 7 is now provided for readers convenience

- Control Cryogel
- Cryogel with GAG
- Cryogel with VEGF
- Cryogel with GAG and VEGF
- Cryogel with FGF2
- Cryogel with VEGF and FGF2
- Cryogel with GAG and FGF2
- Cryogel with GAG, VEGF and FGF2

9. In supplementary figure 2, 3 and 4, the components of 'naïve group' is not clear.

The naming has been updated from naïve to control for a clearer understanding

10. The language used in the paper needs to be significantly improved to increase clarity. In addition, the entire manuscript needs to be thoroughly reviewed for spellings and typographical errors.

The language has been checked and corrected by multiple people and one which have lived in a native English-speaking country for several years

Reviewer #3 (Remarks to the Author):

Manuscript Number: COMMSBIO 22 1236 T

The authors investigated the potential of chemoenzymatically synthesized glycosaminoglycans (GAGs) for controlled binding and release of growth factors. After screening a library of 52 GAGs, selected synthetic GAGs were used to functionalize nanoporous or macroporous hydrogels based on elastin-like recombinamers (ELRs). These materials were loaded with pro-angiogenic growth factors to direct cellular response in a controlled manner. As a proof-of-concept (a) in vitro the tubular morphogenesis of endothelial cells seeded on hydrogels and (b) in vivo the tissue response upon subcutaneous implantation of macroporous hydrogels into a mouse model were studied.

The idea and general concept of incorporating synthetic GAGs into hydrogels to modulate growth factor concentration is interesting and would overcome many problems of hydrogels based on naturally derived GAGs that are highly heterogeneous

[doi.org/10.1021/acs.accounts.9b00420]. However, the authors should nevertheless mention in the manuscript that naturally derived unmodified and modified GAGs have already been successfully used for many years in hydrogels for such purposes - relevant papers should be cited (please also compare Specific comments). It has also been shown for naturally derived GAGs (often heparin) that by varying the degree of sulfation of the GAG and/or the GAG concentration in the hydrogel, uptake and release of signalling proteins can be controlled (e.g., doi.org/10.1016/j.biomaterials.2018.07.056, doi: 10.1002/ADFM.202000068, doi: 10.1039/C9FD00016J).

Macroporous GAG-based hydrogels were also frequently used for sustained protein delivery (e.g., doi: 10.1016/J.BIOMATERIALS.2021.121170), for instance to induce vascularization like in the paper under consideration (doi: 10.1016/j.actbio.2014.11.002, doi: 10.1002/mabi.202100234 or doi: 10.1021/acs.biomac.8b00331). This "State of the Art" should be added to the manuscript.

Another major criticism is that the authors do not show whether the tunability of binding and release of growth factors by the synthetic GAGs that they found for surface-bound GAGs is maintained when the GAGs are incorporated into hydrogel networks and whether similar trends are found here as for the surface-bound GAGs. As only a single GAG was selected to produce the hydrogel materials used for the in vitro and in vivo experiments, the advantages and a convincing evidence of the modularity of a platform based on different synthetic GAG-based hydrogels are not really shown experimentally. Regarding the ELR component of the hydrogels the authors mention in the introduction that "Due to the recombinant expression, several modifications can be made, such as introducing a RGD motif for attachment, a matrix metalloproteinase domain for controlled degradation or functionalization of lysine groups for further modification." (page 5, lines 98-100), but later do not further explain the properties of the specific ELR that they used in their study (see below). Here, too, it would have made sense to compare at least two different ELRs to really demonstrate the modularity of the material system.

The Methods part should please be improved.

Some calculations and interpretations are not comprehensible to me or wrong in my opinion (details see below).

Specific comments:

1. The title could have been chosen more appropriately to emphasize that this study also reports about biomaterial development. Moreover, it is not the elastin-like recombinamers that are macroporous, but the hydrogels made from them.

The title has been updated from:

Synthetic glycosaminoglycans and macroporous elastin-like recombinamers as a modular platform for tissue regeneration

To: (Line 1-2)

Controlled release of growth factors using synthetic glycosaminoglycans in a modular macroporous scaffold for tissue regeneration

2. The selection of the abbreviations listed and explained seems arbitrary. Why, for example, is only the abbreviation of some growth factors explained?

The abbreviations list has been updated to be more extensive. (Line 23-75)

3. The wording of the abstract could be more precise. In particular, I would also mention that K_d values were determined.

I suggest naming the technique used to introduce the macropores not as cryogelation; it is rather a combination of porogen leaching and cryogelation.

In the abstract the authors do not mention that they also tested VEGF loaded macroporous hydrogels for the in vivo studies.

The abstract has been updated to include these suggestions but there was also a need to shorten it to meet the format guidelines of approximate 150 words: (Line 77-90)

Healthy regeneration of tissue relies on a well-orchestrated release of growth factors. Herein, we show the use of synthetic glycosaminoglycans for controlled binding and release of growth factors to induce a desired cellular response. First, we screened glycosaminoglycans with growth factors of interest to define K_{on} (association rate constant), K_{off} (dissociation rate constant), and K_d (equilibrium rate constant). As proof-of-concept, we functionalized an elastin-like recombinamer (ELR) hydrogel with a synthetic glycosaminoglycan and immobilized fibroblast growth factor 2 (FGF2) and/or vascular endothelial growth factor (VEGF), demonstrating that cultured human umbilical vein endothelial cells differentiated into tube-like structures. Taking this concept further, we developed a tunable macroporous ELR cryogel, containing the synthetic glycosaminoglycan and FGF2 that showed increased blood vessel formation and reduced immune response compared to control when implanted in a subcutaneous mouse model. These results demonstrated the possibility for specific release of desired growth factors in a modular structure in vitro and in vivo.

4. The introduction could be more conclusive and include more references to the current "State of the Art". The authors mention only few examples of how in general sustained

release from biomaterials can be achieved. More strategies reported in original articles and review articles could be added here.

For instance, the review article doi: 10.1039/C3TB20853B summarizes “Chemical strategies for the presentation and delivery of growth factors”. The strategy pursued in nature of adjusting growth factor gradients via GAGs, which the authors describe as very advantageous, has also been used in the design of biomaterials to modulate growth factor concentrations and is in principle not new (doi:10.3389/fcell.2021.760532).

The authors should mention this and cite some relevant previous studies, e.g. doi: 10.1016/j.actbio.2014.11.002, doi: 10.1002/mabi.202100234, doi: 10.1002/ADFM.202000068, doi: 10.1021/acs.biomac.8b00331 or doi: 10.1016/J.BIOMATERIALS.2021.121170, to name only some examples of heparin-based hydrogels.

From the introduction in its present form the exact intention of the study is not clear. It is argued somewhat arbitrarily with various aspects of ECM, which are apparently to be imitated by the materials presented here. Why both nanoporous and macroporous systems were investigated is not plausible for me. In part, the impression is given that macroporous hydrogels are generally better suited for tissue regeneration and that their too fast release of growth factors is a problem that should be solved using principles of nature, i.e. by creating GAG-based systems. A specific comparison of the introduced materials with and a differentiation from already established GAG-based hydrogel systems is missing.

Some explanations are not coherent from my point of view: Page 4, line 70/71: „... by having growth factors incorporated into scaffolds such as hydrogels, where the release is limited due to the low area to volume ratio.”

Thank you, the suggested articles are indeed highly relevant and have now been added to the introduction. (Line 112-124)

5. Characterization of GAG-growth factor interactions:

a. Although precise information on the sequences of the various synthetic GAGs can be found in the SI, it would be helpful for the reader to mention some information on the GAGs that could play a role in growth factor binding directly in the manuscript, e.g. as a table summarizing the degree of sulfation, the molar mass, the number of negatively charged groups (e.g. also carboxyl groups that could mediate the binding of growth factors to non-sulfated GAGs, such as GAG no. 3, see p. 6, line 130/13). So far, only the numbers assigned to the GAGs are given here, or the sequences of the selected GAGs are shown very small in the axis in Figure 1. In addition, the specification of important properties of the growth factors used, such as molar mass, IEP, dimensions, would help to better interpret/understand the interaction with the GAGs. The analysis should therefore be based more on physicochemical properties of the components and be comprehensible to the reader through a systematic presentation. In particular, a comparison with the only natural GAG heparin (no. 1) would perhaps also be interesting here (or at least a hint that heparin is hidden behind no. 1), whereby knowledge of the molar mass (distribution) used would also

be interesting here and a comparison with literature data (e.g. with doi:10.3390/molecules24183360).

For easier understanding of the data, a table for the GAGs (Table 1) and the growth factors (Table 2) have been added to the methods section with the data mentioned above. The data is also compared to the mentioned literature: 10.3390/molecules24183360.

b. On page 6, lines 114-117, the authors state "The growth factors added to the microarray revealed three groups of glycosaminoglycans: those that showed high binding to all growth factors, those that showed low/no binding and finally those that showed unique binding depending on the growth factor. For group 2, information is given in the legend to Figure 1. For the other two groups, corresponding information should be please given as well. One possibility could be to include it in the additional table on the GAG characteristics.

To clarify, the text has been changed as shown below and a supplement figure has been added for the microarray screening of additional growth factors

From:

Fig. 1 A-C show vascular endothelial growth factor (VEGF) and hepatocyte growth factor (HGF) as an example of the results obtained with this method. The growth factors added to the microarray revealed three groups of glycosaminoglycans: those that showed high binding to all growth factors, those that showed low/no binding and finally those that showed unique binding depending on the growth factor.

To further study this on a kinetic level, we decided to evaluate a subset of the glycosaminoglycans (one low/no binding, GAG nr 3, one high binding, GAG nr 19, and four with unique binding, GAG nr 10, 26, 34 and 43) using surface plasmon resonance (SPR) (see Supplement Table 2).

To: (Line 151-164)

Fig. 1 A-C show vascular endothelial growth factor (VEGF) and hepatocyte growth factor (HGF), furthermore, BMP4, CXCL12, IL-6, FGF2, PDGF-AA, KGF and TGF beta 1 were also tested and data can be found in Supplement Figure 1. The growth factors added to the microarray revealed three groups of glycosaminoglycans: those that showed high binding to all growth factors, those that showed low/no binding and finally those that showed binding depending on the growth factor.

To further study this on a kinetic level, we decided to evaluate a subset of the glycosaminoglycans using surface plasmon resonance (SPR). One glycosaminoglycan with low to no binding to any growth factors, GAG nr 3, one high binding (except for TGF beta 1), GAG nr 19, and four with binding to only a subset of the growth factors, GAG nr 10, 26, 34 and 43 (see Supplement Table 2).

c. In addition, there is a certain discrepancy between the number of GAGs examined in the text (52), in the table (nr. 1-53=54) and in Figure 1 (nr. 0 - 53=54). A clear allocation of the GAGs in the figures is therefore not possible.

For a clearer and easier understanding, the text has been updated from:

To demonstrate the possibility to use synthetic glycosaminoglycans as a tool for customizing the release rate of growth factors, a range of them were screened on a microarray functionalized with 52 different synthetic glycosaminoglycans (see Supplement Table 1).

To: (Line 147-151)

To demonstrate the possibility to use synthetic glycosaminoglycans as a tool for customizing the release rate of growth factors, a range of them were screened on a microarray functionalized with 52 different synthetic glycosaminoglycans, referred to as 2-53 while 1 is commercially available heparin and, when applicable, 0 is background values (see Supplement Table 1).

d. Is the use of equal mass concentrations of the different growth factors useful in microarray screening, as is the use of equal mass concentrations of the antibodies? Would not an equal molar concentration of substances make more sense?

As there is no way of knowing the number of different binding sites of a glycosaminoglycan and if they are competing for the same binding site as the antibody, the microarray was used as a way to screen factors and thus a very high concentration was used. For a deeper understanding of the binding properties, the glycosaminoglycans were further evaluated using SPR.

What protein concentrations were used for SPR? I could not find any information on this in the manuscript.

The concentrations used have been added to the methods section and can be seen in Table 2

e. The formula given on page 25 is not correct in my opinion. It should correctly read $K_d = K_{off}/K_{on}$.

Thank you for pinpointing this issue. We have corrected it in the revised version of the manuscript.

Furthermore, the authors should definitely check all the values for K_d given in the SI (12692_0_data_set_98338_r3frs0.xlsx). I calculated different values except for the two growth factors CXCL12 and CXCL4. This would also change the presentation in the Results or Discussion Part in some cases. For example, for GAG nr. 19 I do not calculate the same K_d for VEGF (410 nM) and (FGF-2 (413 nM)), as the authors state and interpret on page 7, line 137 and page 20/21, lines 397, but a 3 times higher K_d for VEGF (136 nM) than for FGF-2 (51 nM) underpinning the higher effect for FGF-2 found in the cell experiment. It is questionable whether the statement that "focus should be on release rate and not total binding when looking to sustain the efficacy of growth factors when using synthetic glycosaminoglycans" can still be sufficiently substantiated.

Thank you very much for going into the supplement and finding this mistake. Due to an Excel error, the K_d values were not correctly calculated from K_{on} and K_{off} , Figure 1 has been updated and shows the corrected values, as can be seen below, as well as in supplement table 2.

From:

To:

f. Due to the incorrect calculation of the Kd values, the factor 2000 on page 18, line 340 ff, is incorrect in my opinion, too. The correct value would be about 1000.

This value has been updated in the article.

g. The claim made on page 6, lines 124-127 "The synthetic glycosaminoglycan that had the longest carbohydrate chain as well as the highest degree of sulfation, GAG nr 19 (highlighted in orange in Fig. 1E), showed unique binding to transforming growth factor beta 1 (TGF beta 1), keratinocyte growth factor (KGF) and connective tissue growth factor (CCN2)". I cannot understand this on the basis of Figure 1. Is it more likely that the following is meant: "The synthetic glycosaminoglycan with the longest carbohydrate chain as well as the highest degree of sulphation, GAG No. 19 (highlighted in orange in Fig. 1E), was the only one to show binding to transforming growth factor beta 1 (TGF beta 1), keratinocyte growth factor (KGF) and connective tissue growth factor (CCN2)"?

Thank you. The text has been updated with your suggestion for an easier understanding.

6. Production of the hydro-/cryogel materials:

a. The description of hydro- and cryogel production should be optimized. In the current way it is not understandable and not comprehensible.

The method part has been optimized. For example this section:

Water was sprayed using on top of liquid nitrogen and sieved through strainers with mesh sizes 500, 200, and 100 to collect ice crystals of desired size. Ice crystals were added to a mold together with the dissolved ELR and left in a cryostat at -5 °C to equilibrate.

Have been changed to: (Line: 624 -629)

MilliQ water (Millipore Reference, Merck) was sprayed using a spray bottle on top of liquid nitrogen and sieved through strainers with mesh sizes 500, 200, and 100 µm (Pluristrainer, 43-50500-03, 43-50200-03, 43-50100-51, pluriSelect Life Science) to collect ice crystals of desired sizes. Ice crystals were added to a mold (here a syringe was used (1ml syringe, 329654, BD Bioscience)). Ice crystal and dissolved ELR were left in a cryostat (HM500M, Microm) at -5° C to equilibrate.

b. It should be explained very briefly how the ELRs are produced, even if this work has already been published. I would recommend including a brief description of the properties (e.g., molecular weight, RGD sequences, functionalisation with cyclooctyne and azide).

This has been described in the methods section, and a table with the ELR sequences and molecular weights has been included (Table 3). The following text has been added to the manuscript: (Line:586-603)

The design of the ELRs used in this work, termed HE5 and HRGD6, has been described elsewhere (26), and table 3 includes their sequences and molecular weights. Each one comprises different bioactive domains. HE5 contains matrix metalloproteinase (MMP)-

sensitive domains that allow biodegradation mainly by MMP-2, MMP-9 and MMP-13 (26) On the other hand, HRGD6 include Arg-Gly-Asp (RGD) for cell attachment and interaction, improving the biocompatibility of the scaffolds made with it, like the macroporous cryogels (51).

Both ELRs were produced as previously described(52) . In brief, the ELR genes were cloned into a pET-25b(+) plasmid (Novagen, Merck) for expression in *Escherichia coli* (BLR(DE3) strain, Novagen, Merck) in a 15-L bioreactor (Applikon Biotechnology). The ELRs were subsequently purified through consecutive heating and cooling steps, dialyzed against ultra-pure water and filtered through a 0.22 µm filters (Nalgene) prior to freeze-drying.

c. Instead of using mass concentrations and ratios, it would be probably better to work with molar quantities in some cases.

The concentrations and ratios have been given as in previous publications. Thus, to make our publication comparable to those, we would prefer to keep this format for the quantities. However, we have provided the information needed for the calculation of the molar quantities, if needed.

d. It is not really clear to me which functional groups are involved in the hydrogel formation and functionalization, especially in the incorporation of growth factors and GAGs, and what the molar ratios of ELR : growth factor : GAG are.

Thank you for this comment, all the data is now provided for the molar quantities. On line 615-616 it can be seen that HE5-C has 3-4 lysines modified with cyclooctyne and HRGF6-N3 has 12-14 lysines modified with Azide. Each synthetic glycosaminoglycan has a backbone with one azide.

Is the growth factor added during the production of the hydrogel materials or are the materials loaded with the proteins afterwards? Apparently, this is handled differently for the hydrogels and the cryogels.

The growth factors for the hydrogel evaluation are added during production but as the liquid is sublimated during freeze-drying for the cryogel production, it is then added afterwards.

e. The cross-linking chemistry for the different systems should be please explained more clearly in a scheme. So far, single important information is given in the Results, Discussion and Methods Part and the reader need somehow to combine it.

The modification and cross-linking chemistry have been now described in the methods section of the article as follows: (Line: 604-612)

The chemical modification of the ELRs was performed following well established methods, as described by Gonzalez de Torre et al.(53) . The aim was to enable a strain-promoted azide alkyne cycloaddition (SPAAC) 'click chemistry' reaction (54) in order to achieve the cross-linking of two different ELRs to form a network that gives a stable scaffold (hydrogel or

cryogel). To this end, azide (N_3) groups were introduced through the modification of the free ϵ -amine of the lysine residues present in the HRGD6 ELR, giving HRGD6- N_3 . Similarly, cyclooctyne groups were conjugated to the HE5, to render a HE5-C. Both modified ELRs were used for the SPAAC reaction and hydrogel/cryogel formation. Table 4 shows the modification degree obtained for the different ELRs.

Further information and a scheme can be found in the seminal article about this cross-linking approach using ELRs (also referenced in the manuscript):
<https://www.sciencedirect.com/science/article/pii/S1742706114000580>

7. Characterization of the hydro/cryogel materials:

a. So far the authors present the mechanical characterization of only the pure ELR cryogels with different pore sizes (two methods were used to determine storage modulus and Young's modulus; the values obtained are not compared or discussed further). A comparison with GAG-containing cryogels would have been useful to see if/how the GAG changes the mechanical properties of the materials.

The added GAGs will bind to approximate 1/500 of the total binding sites available for cross linkage, thus having a minimal impact on the overall cross linkage amount.

b. Also, the pore structure is only shown for the pure ELR cryogels and not for the GAG-containing ones, which are then ultimately used in the in vivo experiment.

If it is not known whether the pore structure and the mechanical properties are the same as in the control, it cannot be ruled out that changes in the cell response also result from a changed overall or local stiffness (which is not determined at all) or pore geometry and not only from the release of the growth factor.

Please see the new supplement figure 2 A and B where A shows the pore size without the added GAGs and B shows pore size with GAGs added demonstrating no observable difference

A) With No GAGs

B) With GAGs

c. It is a nice result that macroporous hydrogels (cryogels) with different pore sizes can be produced, but the influence of the pore size on the in vivo response is not investigated (cf. also above, influence of the GAG component).

Thank you for this comment, for more in depth evaluation on how pore size influences the in vivo response we recommend the following articles [10.1016/j.actbio.2013.12.042](https://doi.org/10.1016/j.actbio.2013.12.042) and [10.1038/s41467-019-11397-1](https://doi.org/10.1038/s41467-019-11397-1)

d. For the sake of completeness, the mechanical properties of the ELR and ELR-GAG hydrogels should also be determined and shown.

We thank the reviewer for the comment, which we have addressed for the ELR hydrogels. The results obtained can be found as Supplementary figures in the new version of the manuscript, and described in the results section of the manuscript, as follows: (Line 255-258, and 652-687)

On the other hand, the ELR hydrogel gave a tangential E value of 4041.6 ± 325.8 Pa

(Supplementary Figure 2), while the G' was found to be 633.8 ± 95.9 Pa (see Supplementary Figure 3 for the strain sweep curve). Both measurements resulted in a much higher stiffness than for all the groups of the cryogel counterparts.

The methods and discussion sections have also been updated to include the mechanical measurements of the ELR hydrogels.

As regards GAG-modified hydrogels, they are not expected to show changes in the mechanical properties compared to the unmodified ones, considering the low degree of conjugation (approximately 1 out of 500 ELR molecules modified with the GAG) and previous works where a low modification has not led to significant changes in hydrogel morphology nor mechanical properties (e.g., <https://www.sciencedirect.com/science/article/pii/S174270612100372X> or <https://link.springer.com/article/10.1007/s10856-019-6232-z>).

e. In my opinion, the most important point missing regarding the characterization of the ELR and ELR-GAG hydrogel materials to substantiate the study and make it convincing are studies on the uptake and release of the growth factors in the GAG-modified hydrogels and cryogels. The binding studies on the surface-bound GAGs by means of the screening assay and SPR can certainly only provide initial indications of the binding and release behavior of various growth factors to GAG-containing hydrogel materials, but cannot be used alone to interpret the in vitro and in vivo data shown and cell-instructive properties of the hydrogel materials in general. It is expected that the comparison of uptake and release of VEGF and FGF-2 from GAG nr. 19-containing cryogels will provide important clues to better understand the observed in vivo response.

This question is of course of great interest, and we have therefore tried to answer this by setting up an experiment where we loaded VEGF into two groups of cryogels, one made with GAGs and one without. We had a lot of problems with too high and too low signals, probably due to VEGF sticking to the plastic in tubes and pipette tips. After some optimization we were able to produce the graph below but unfortunately the cryogels with GAGs showed too high signal (as can be seen with the values being the absorbance maximum of around 4) and cryogels with no GAGs had technical difficulties. The graph indicates a release of VEGF from the cryogels functionalized with GAGs a half time of around 3h. Compared to our SPR data for GAG and VEGF, a half time of around 20 seconds was measured. If this data is representative, the bulk release from the cryogel was around 500 times slower compared the GAG-VEGF complex alone. That the bulk release would be slower is often the case as a VEGF molecule can jump from one GAG to another, thus stays in the cryogel for a longer time.

f. Why is the combination of FGF-2 with additional VEGF only investigated in the cryogels/in vivo and not in the hydrogels/in vitro experiments?

The main aim of our modular system was to show its usability for tissue regeneration in vivo and how it can be used in vitro to control cell activity in a completely defined system. In vitro, the access of cells was a limitation and to show proof-of-concept in this setting we chose just to optimize for FGF2. In vivo, according to literature the formation of blood vessels is known to require both the activation of VEGF and FGF2, which is why we chose this combination.

g. The addition of VEGF showed no significant effect in the in vivo experiment. However, only 1/5 of the mass of FGF-2 was used to load the cryogels. Furthermore, the loading efficiency may be different for the two factors. However, this was not determined. The analysis of the release behavior of the two different factors from the cryogel materials could be more helpful in the interpretation of the data than the comparison of the Kd values determined on surface-bound GAGs.

The values used were taken from this very nice article, [10.1242/jcs.02483](https://doi.org/10.1242/jcs.02483), as too much growth factors delivered also gives adverse effects.

h. Did the authors check whether autoclavation changes the properties of the cryogels?

The cryogels did shrink to an extent during autoclaving but similar to the non-autoclaved ones they recover their size when hydrated

8. The authors should explain why they use a hydrogel for the in vitro experiments and a macroporous cryogel for the in vivo experiments. Why is the morphogenesis of HUVECs studied in 2D and not in 3D in the hydrogel system (especially since the abstract also mentions 3D in vitro systems)?

As a proof of principle, to show that synthetic glycosaminoglycans linked to the ELR gel were able to interact with cell surface receptors also after immobilization in the hydrogel, we chose an assay with an easy and clear readout and therefore we chose the HUVEC tube formation assay. This is a 2D assay and the abstract has been updated to reflect this.

9. The technical terms should be used correctly. Firstly, it is confusing when a capital K is used for both the thermodynamic quantity equilibrium dissociation constant (Kd) and the kinetic constants Kon and Koff. Usually, a small k is used for the kinetic velocity constants. Also, it should correctly read "equilibrium dissociation constant", "dissociation rate constant" and "association rate constant" instead of "total binding" or "equilibrium constant", "dissociation constant" and "association constant". Accordingly, on page 6, line 118, it should also correctly read " ... on a kinetic and thermodynamic level ...".

Thank you for pointing this out. It has been corrected for in the text.

I think the authors do not mean "naïve" but "native" and not "capsel" but "capsule (e.g. Figure S11, I)".

The capsule naming has been updated

The manuscript has been updated and the naming have been changed from “naïve” to “control”

10. Some statistical data are missing, for instance in Figure 1 A and C the authors do not indicate what is plotted here and what is shown by the error bars. Please add to the Figure Legend or add a separate section “Statistics”, where you give this important information.

The figure legend has been updated with the following text:

For A and C data is shown in a box and whisker (as Tukey) of 36 binding spots for each synthetic glycosaminoglycans

Similarly, for the determination of K_d , K_{on} and K_{off} , the number of measurements and the errors are not given (neither in the manuscript nor in the tables in the SI). Was only one experiment carried out?

All growth factors for the SPR experiments were run at 5 different concentrations and the trueness of the fit, χ^2 is available in the supplement for each growth factor.

11. The References should be in a uniform format. Ref. 14 contains two-times the term “Author manuscript”. Sometimes doi nummers are given, sometimes not.

Thank you the manuscript has been updated to use natures referencing style.

12. The labels in the figures are in part clearly too small.

We agree that the labels should be easily readable, and this will be updated when we know the final size for print of each figure.

Reviewers' comments:

Reviewer #1 (Remarks to the Author):

The authors have suitably revised the manuscript, including the addition of new data. I therefore think that the manuscript is suitable for publication.

Reviewer #2 (Remarks to the Author):

The authors have responded to all the comments satisfactorily.

Reviewer #3 (Remarks to the Author):

Reviewers' comments together with answers:

Reviewer #3 (Remarks to the Author):

Reviewer #3: The manuscript has been very much improved. However, in my opinion there are still some points that need to be addressed/considered. Please compare my comments below ("Reviewer #3: ..." inserted after the authors' answers). Figures have been removed.

Manuscript Number: COMMSBIO 22 1236 T

The authors investigated the potential of chemoenzymatically synthesized glycosaminoglycans (GAGs) for controlled binding and release of growth factors. After screening a library of 52 GAGs, selected synthetic GAGs were used to functionalize nanoporous or macroporous hydrogels based on elastin-like recombinamers (ELRs). These materials were loaded with pro-angiogenic growth factors to direct cellular response in a controlled manner. As a proof-of-concept (a) in vitro the tubular morphogenesis of endothelial cells seeded on hydrogels and (b) in vivo the tissue response upon subcutaneous implantation of macroporous hydrogels into a mouse model were studied. The idea and general concept of incorporating synthetic GAGs into hydrogels to modulate growth factor concentration is interesting and would overcome many problems of hydrogels based on naturally derived GAGs that are highly heterogeneous [doi.org/10.1021/acs.accounts.9b00420]. However, the authors should nevertheless mention in the manuscript that naturally derived unmodified and modified GAGs have already been successfully used for many years in hydrogels for such purposes - relevant papers should be cited (please also compare Specific comments). It has also been shown for naturally derived GAGs (often heparin) that by varying the degree of sulfation of the GAG and/or the GAG concentration in the hydrogel, uptake and release of signalling proteins can be controlled (e.g., doi.org/10.1016/j.biomaterials.2018.07.056, doi: 10.1002/ADFM.202000068, doi: 10.1039/C9FD00016J). Macroporous GAG-based hydrogels were also frequently used for sustained protein delivery (e.g., doi: 10.1016/J.BIOMATERIALS.2021.121170), for instance to induce vascularization like in the paper under consideration (doi: 10.1016/j.actbio.2014.11.002, doi: 10.1002/mabi.202100234 or doi: 10.1021/acs.biomac.8b00331). This "State of the Art" should be added to the manuscript. Another major criticism is that the authors do not show whether the tunability of binding and release of growth factors by the synthetic GAGs that they found for surface-bound GAGs is maintained when the GAGs are incorporated into hydrogel networks and whether similar trends are found here as for the surface-bound GAGs. As only a single GAG was selected to produce the hydrogel materials used for the in vitro and in vivo experiments, the advantages and a convincing evidence of the modularity of a platform based on different synthetic GAG-based hydrogels are not really shown experimentally. Regarding the ELR component of the hydrogels the authors mention in the introduction that "Due to the recombinant expression, several modifications can be made, such as introducing a RGD motif for attachment, a matrix metalloproteinase domain for controlled degradation or functionalization of lysine groups for further modification." (page 5, lines 98-100), but later do not further explain the properties of the specific ELR that they used in their study (see below). Here, too, it would have made sense to compare at least two different ELRs to really demonstrate the modularity of the material system.

The Methods part should please be improved.

Some calculations and interpretations are not comprehensible to me or wrong in my opinion (details see below).

Specific comments:

1. The title could have been chosen more appropriately to emphasize that this study also reports about biomaterial development. Moreover, it is not the elastin-like recombinamers that are macroporous, but the hydrogels made from them.

Authors: The title has been updated from:

Synthetic glycosaminoglycans and macroporous elastin-like recombinamers as a modular platform for tissue regeneration

To: (Line 1-2)

Controlled release of growth factors using synthetic glycosaminoglycans in a modular macroporous scaffold for tissue regeneration

Reviewer #3: The updated title fits better. However, a suggestion from my side would be the following title: Macroporous scaffolds based on synthetic glycosaminoglycans and elastin-like recombinamers as a modular platform for tissue regeneration

2. The selection of the abbreviations listed and explained seems arbitrary. Why, for example, is only the abbreviation of some growth factors explained?

Authors: The abbreviations list has been updated to be more extensive. (Line 23-75)

Reviewer #3: ok

3. The wording of the abstract could be more precise. In particular, I would also mention that Kd values were determined.

I suggest naming the technique used to introduce the macropores not as cryogelation; it is rather a combination of porogen leaching and cryogelation.

In the abstract the authors do not mention that they also tested VEGF loaded macroporous hydrogels for the in vivo studies.

Authors: The abstract has been updated to include these suggestions but there was also a need to shorten it to meet the format guidelines of approximate 150 words: (Line 77-90)

Healthy regeneration of tissue relies on a well-orchestrated release of growth factors. Herein, we show the use of synthetic glycosaminoglycans for controlled binding and release of growth factors to induce a desired cellular response. First, we screened glycosaminoglycans with growth factors of interest to define K_{on} (association rate constant), K_{off} (dissociation rate constant), and K_d (equilibrium rate constant). As proof-of-concept, we functionalized an elastin-like recombinamer (ELR) hydrogel with a synthetic glycosaminoglycan and immobilized fibroblast growth factor 2 (FGF2) and/or vascular endothelial growth factor (VEGF), demonstrating that cultured human umbilical vein endothelial cells differentiated into tube-like structures. Taking this concept further, we developed a tunable macroporous ELR cryogel, containing the synthetic glycosaminoglycan and FGF2 that showed increased blood vessel formation and reduced immune response compared to control when implanted in a subcutaneous mouse model. These results demonstrated the possibility for specific release of desired growth factors in a modular structure in vitro and in vivo.

Reviewer #3: I suggest to additionally replace:

... First, we screened glycosaminoglycans with growth factors of interest to define K_{on} (association rate constant), K_{off} (dissociation rate constant), and K_d (equilibrium rate constant). As proof-of-concept, we functionalized an elastin-like recombinamer (ELR) hydrogel with a synthetic glycosaminoglycan and immobilized fibroblast growth factor 2 (FGF2) and/or vascular endothelial growth factor (VEGF), demonstrating that cultured human umbilical vein endothelial cells

differentiated into tube-like structures. Taking this concept further, we developed a tunable macroporous ELR cryogel, containing the synthetic glycosaminoglycan and FGF2 that showed increased blood vessel formation and reduced immune response compared to control when implanted in a subcutaneous mouse model. These results demonstrated the possibility for specific release of desired growth factors in a modular structure in vitro and in vivo.

By:

... First, we screened glycosaminoglycans with growth factors of interest to determine k_{on} (association rate constant), k_{off} (dissociation rate constant), and K_d (equilibrium rate constant). As proof-of-concept, we functionalized an elastin-like recombinamer (ELR) hydrogel with a synthetic glycosaminoglycan and immobilized fibroblast growth factor 2 (FGF2), demonstrating that cultured human umbilical vein endothelial cells on top differentiated into tube-like structures. Taking this concept further, we developed a tunable macroporous ELR cryogel material, containing a synthetic glycosaminoglycan and FGF2 that showed increased blood vessel formation and reduced immune response compared to control when implanted in a subcutaneous mouse model. These results demonstrated the possibility for specific release of desired growth factors in/from a modular 3D scaffold in vitro and in vivo.

(As far as I understood VEGF was not used in the in vitro studies.)

4. The introduction could be more conclusive and include more references to the current "State of the Art". The authors mention only few examples of how in general sustained release from biomaterials can be achieved. More strategies reported in original articles and review articles could be added here.

For instance, the review article doi: 10.1039/C3TB20853B summarizes "Chemical strategies for the presentation and delivery of growth factors". The strategy pursued in nature of adjusting growth factor gradients via GAGs, which the authors describe as very advantageous, has also been used in the design of biomaterials to modulate growth factor concentrations and is in principle not new (doi:10.3389/fcell.2021.760532).

The authors should mention this and cite some relevant previous studies, e.g. doi: 10.1016/j.actbio.2014.11.002, doi: 10.1002/mabi.202100234, doi: 10.1002/ADFM.202000068, doi: 10.1021/acs.biomac.8b00331 or doi: 10.1016/J.BIOMATERIALS.2021.121170, to name only some examples of heparin-based hydrogels.

From the introduction in its present form the exact intention of the study is not clear. It is argued somewhat arbitrarily with various aspects of ECM, which are apparently to be imitated by the materials presented here. Why both nanoporous and macroporous systems were investigated is not plausible for me. In part, the impression is given that macroporous hydrogels are generally better suited for tissue regeneration and that their too fast release of growth factors is a problem that should be solved using principles of nature, i.e. by creating GAG-based systems. A specific comparison of the introduced materials with and a differentiation from already established GAG-based hydrogel systems is missing.

Some explanations are not coherent from my point of view: Page 4, line 70/71: "... by having growth factors incorporated into scaffolds such as hydrogels, where the release is limited due to the low area to volume ratio."

Authors: Thank you, the suggested articles are indeed highly relevant and have now been added to the introduction. (Line 112-124)

Reviewer #3: The authors added some relevant references to the introduction. However, some aspects already mentioned in the general part of my first review were not considered in this context. In particular, in my opinion the following text parts have to be revised: (line 108 -122) "The disadvantage of manufactured macroporous scaffolds incorporated with growth factors is quick release due to high exposure area. To circumvent the fast release from macroporous structures, different strategies have been designed¹¹; using aptamers¹², or creating

a binding tag 12,13 . However, these approaches only allow for either binding of one single factor or only the loaded factors and not binding factors produced in the body. ... This, in combination with the lack of sequencing technologies, have led to the use of long undefined glycosaminoglycans purified from different origins^{16–20} , that bind growth factors but lack the possibility of fine tuning release properties ²¹ .”

As already mentioned in the general first introducing paragraph of my original comments, it has also been shown for hydrogels based on naturally derived GAGs (often heparin) that by varying the degree of sulfation of the GAG and/or the GAG concentration in the hydrogel, uptake and release of signalling proteins can be controlled (e.g., doi.org/10.1016/j.biomaterials.2018.07.056, doi: 10.1002/ADFM.202000068, doi: 10.1039/C9FD00016J). The same basic concept also enabled the development of macroporous GAG-based hydrogels for sustained and fine-tuned protein delivery (e.g., doi: 10.1016/J.BIOMATERIALS.2021.121170) (thus, these materials do not “lack the possibility of fine tuning release properties” or have the disadvantage of “quick release due to high exposure area” as stated by the authors).

5. Characterization of GAG-growth factor interactions:

a. Although precise information on the sequences of the various synthetic GAGs can be found in the SI, it would be helpful for the reader to mention some information on the GAGs that could play a role in growth factor binding directly in the manuscript, e.g. as a table summarizing the degree of sulfation, the molar mass, the number of negatively charged groups (e.g. also carboxyl groups that could mediate the binding of growth factors to non-sulfated GAGs, such as GAG no. 3, see p. 6, line 130/13). So far, only the numbers assigned to the GAGs are given here, or the sequences of the selected GAGs are shown very small in the axis in Figure 1. In addition, the specification of important properties of the growth factors used, such as molar mass, IEP, dimensions, would help to better interpret/understand the interaction with the GAGs. The analysis should therefore be based more on physicochemical properties of the components and be comprehensible to the reader through a systematic presentation. In particular, a comparison with the only natural GAG heparin (no. 1) would perhaps also be interesting here (or at least a hint that heparin is hidden behind no. 1), whereby knowledge of the molar mass (distribution) used would also be interesting here and a comparison with literature data (e.g. with doi:10.3390/molecules24183360).

Authors: For easier understanding of the data, a table for the GAGs (Table 1) and the growth factors (Table 2) have been added to the methods section with the data mentioned above. The data is also compared to the mentioned literature: 10.3390/molecules24183360.

Reviewer #3: The added tables are very helpful for the reader. Unfortunately, the units in Table 1 are missing and should be added by the authors. Moreover, they should explain, what they exactly mean by “number of sulphation ´s” and “sugar length”.

b. On page 6, lines 114-117, the authors state "The growth factors added to the microarray revealed three groups of glycosaminoglycans: those that showed high binding to all growth factors, those that showed low/no binding and finally those that showed unique binding depending on the growth factor. For group 2, information is given in the legend to Figure 1. For the other two groups, corresponding information should be please given as well. One possibility could be to include it in the additional table on the GAG characteristics.

Authors: To clarify, the text has been changed as shown below and a supplement figure has been added for the microarray screening of additional growth factors

From:

Fig. 1 A-C show vascular endothelial growth factor (VEGF) and hepatocyte growth factor (HGF) as an example of the results obtained with this method. The growth factors added to the microarray revealed three groups of glycosaminoglycans: those that showed high binding to all growth factors, those that showed low/no binding and finally those that showed unique binding depending on the growth factor.

To further study this on a kinetic level, we decided to evaluate a subset of the glycosaminoglycans (one low/no binding, GAG nr 3, one high binding, GAG nr 19, and four with unique binding, GAG nr

10, 26, 34 and 43) using surface plasmon resonance (SPR) (see Supplement Table 2).

To: (Line 151-164)

Fig. 1 A-C show vascular endothelial growth factor (VEGF) and hepatocyte growth factor (HGF), furthermore, BMP4, CXCL12, IL-6, FGF2, PDGF-AA, KGF and TGF beta 1 were also tested and data can be found in Supplement Figure 1. The growth factors added to the microarray revealed three groups of glycosaminoglycans: those that showed high binding to all growth factors, those that showed low/no binding and finally those that showed binding depending on the growth factor. To further study this on a kinetic level, we decided to evaluate a subset of the glycosaminoglycans using surface plasmon resonance (SPR). One glycosaminoglycan with low to no binding to any growth factors, GAG nr 3, one high binding (except for TGF beta 1), GAG nr 19, and four with binding to only a subset of the growth factors, GAG nr 10, 26, 34 and 43 (see Supplement Table 2).

Reviewer #3: The additional information and changes very much improved this part of the manuscript. : I recommend to replace:

"... those that showed high binding to all growth factors, those that showed low/no binding and finally those that showed binding depending on the growth factor."

By:

"... those that showed high binding to all growth factors, those that showed low/no binding and finally those that showed selective binding depending on the growth factor."

c. In addition, there is a certain discrepancy between the number of GAGs examined in the text (52), in the table (nr. 1-53=54) and in Figure 1 (nr. 0 - 53=54). A clear allocation of the GAGs in the figures is therefore not possible.

Authors: For a clearer and easier understanding, the text has been updated from:

To demonstrate the possibility to use synthetic glycosaminoglycans as a tool for customizing the release rate of growth factors, a range of them were screened on a microarray functionalized with 52 different synthetic glycosaminoglycans (see Supplement Table 1).

To: (Line 147-151)

To demonstrate the possibility to use synthetic glycosaminoglycans as a tool for customizing the release rate of growth factors, a range of them were screened on a microarray functionalized with 52 different synthetic glycosaminoglycans, referred to as 2-53 while 1 is commercially available heparin and, when applicable, 0 is background values (see Supplement Table 1).

Reviewer #3: The text is much clearer now. I suggest to additionally replace:

... and, when applicable, 0 is background values (see Supplement Table 1).

By:

... and, when applicable, the background value (0) is given (see Supplement Table 1).

d. Is the use of equal mass concentrations of the different growth factors useful in microarray screening, as is the use of equal mass concentrations of the antibodies? Would not an equal molar concentration of substances make more sense?

Authors: As there is no way of knowing the number of different binding sites of a glycosaminoglycan and if they are competing for the same binding site as the antibody, the microarray was used as a way to screen factors and thus a very high concentration was used. For a deeper understanding of the binding properties, the glycosaminoglycans were further evaluated using SPR.

What protein concentrations were used for SPR? I could not find any information on this in the manuscript.

Authors: The concentrations used have been added to the methods section and can be seen in Table 2

Reviewer #3: ok

e. The formula given on page 25 is not correct in my opinion. It should correctly read $K_d = K_{off}/K_{on}$.

Authors: Thank you for pinpointing this issue. We have corrected it in the revised version of the manuscript.

Reviewer #3: Ok, however, it is common to use a small k (and not a capital K) for the kinetic velocity constants (thus: kon and koff). The authors please should change it accordingly.

Furthermore, the authors should definitely check all the values for K_d given in the SI (12692_0_data_set_98338_r3frs0.xlsx). I calculated different values except for the two growth factors CXCL12 and CXCL4. This would also change the presentation in the Results or Discussion Part in some cases. For example, for GAG nr. 19 I do not calculate the same K_d for VEGF (410 nM) and FGF-2 (413 nM), as the authors state and interpret on page 7, line 137 and page 20/21, lines 397, but a 3 times higher K_d for VEGF (136 nM) than for FGF-2 (51 nM) underpinning the higher effect for FGF-2 found in the cell experiment. It is questionable whether the statement that "focus should be on release rate and not total binding when looking to sustain the efficacy of growth factors when using synthetic glycosaminoglycans" can still be sufficiently substantiated.

Authors: Thank you very much for going into the supplement and finding this mistake. Due to an Excel error, the K_d values were not correctly calculated from K_{on} and K_{off} , Figure 1 has been updated and shows the corrected values, as can be seen below, as well as in supplement table 2.

Reviewer #3: The mentioned error was corrected in the results part. However, the discussion is still based on the wrong values for the K_d , i.e. equal values for VEGF and FGF-2. The K_d for VEGF is much higher! It is questionable whether the statement that "focus should be on release rate and not total binding when looking to sustain the efficacy of growth factors when using synthetic glycosaminoglycans" can still be sufficiently substantiated.

In my opinion the following paragraph need to be corrected accordingly (otherwise results part and discussion are inconsistent), lines 444-451:

"However, we saw no added effect of including VEGF in the macroporous ELR, when combining it with synthetic glycosaminoglycans. We suggest that this was due to the 100 times faster release (K_{off}) of VEGF compared to FGF2 for the selected synthetic glycosaminoglycan, despite virtually having the same equilibrium dissociation constant (K_d). This indicates in turn that focus should be on K_{off} and not on K_d when looking to sustain the efficacy of growth factors when using synthetic glycosaminoglycans."

From:

..

To:

..

f. Due to the incorrect calculation of the K_d values, the factor 2000 on page 18, line 340 ff, is incorrect in my opinion, too. The correct value would be about 1000.

Authors: This value has been updated in the article.

Reviewer #3: ok

g. The claim made on page 6, lines 124-127 "The synthetic glycosaminoglycan that had the longest carbohydrate chain as well as the highest degree of sulfation, GAG nr 19 (highlighted in orange in Fig. 1E), showed unique binding to transforming growth factor beta 1 (TGF beta 1), keratinocyte growth factor (KGF) and connective tissue growth factor (CCN2)". I cannot understand this on the basis of Figure 1. Is it more likely that the following is meant: "The synthetic glycosaminoglycan with the longest carbohydrate chain as well as the highest degree of sulfation, GAG No. 19 (highlighted in orange in Fig. 1E), was the only one to show binding to

transforming growth factor beta 1 (TGF beta 1), keratinocyte growth factor (KGF) and connective tissue growth factor (CCN2)"?

Authors: Thank you. The text has been updated with your suggestion for an easier understanding.

Reviewer #3: ok

6. Production of the hydro-/cryogel materials:

a. The description of hydro- and cryogel production should be optimized. In the current way it is not understandable and not comprehensible.

Authors: The method part has been optimized. For example this section:

Water was sprayed using on top of liquid nitrogen and sieved through strainers with mesh sizes 500, 200, and 100 to collect ice crystals of desired size. Ice crystals were added to a mold together with the dissolved ELR and left in a cryostat at -5 °C to equilibrate.

Have been changed to: (Line: 624 -629)

MilliQ water (Millipore Reference, Merck) was sprayed using a spray bottle on top of liquid nitrogen and sieved through strainers with mesh sizes 500, 200, and 100 μm (Pluristrainer, 43-50500-03 ,43-50200-03, 43-50100-51, pluriSelect Life Science) to collect ice crystals of desired sizes. Ice crystals were added to a mold (here a syringe was used (1ml syringe, 329654, BD Bioscience)). Ice crystal and dissolved ELR were left in a cryostat (HM500M, Microm) at -5° C to equilibrate.

Reviewer #3: The description is more detailed and comprehensible now.

b. It should be explained very briefly how the ELRs are produced, even if this work has already been published. I would recommend including a brief description of the properties (e.g., molecular weight, RGD sequences, functionalisation with cyclooctyne and azide).

Authors: This has been described in the methods section, and a table with the ELR sequences and molecular weights has been included (Table 3). The following text has been added to the manuscript: (Line:586-603)

The design of the ELRs used in this work, termed HE5 and HRGD6, has been described elsewhere (26), and table 3 includes their sequences and molecular weights. Each one comprises different bioactive domains. HE5 contains matrix metalloproteinase (MMP)-sensitive domains that allow biodegradation mainly by MMP-2, MMP-9 and MMP-13 (26) On the other hand, HRGD6 include Arg-Gly-Asp (RGD) for cell attachment and interaction, improving the biocompatibility of the scaffolds made with it, like the macroporous cryogels (51).

Both ELRs were produced as previously described(52) . In brief, the ELR genes were cloned into a pET-25b(+) plasmid (Novagen, Merck) for expression in Escherichia coli (BLR(DE3) strain, Novagen, Merck) in a 15-L bioreactor (Applikon Biotechnology). The ELRs were subsequently purified through consecutive heating and cooling steps, dialyzed against ultra-pure water and filtered through a 0.22 μm filters (Nalgene) prior to freeze-drying.

Reviewer #3: Ok.

c. Instead of using mass concentrations and ratios, it would be probably better to work with molar quantities in some cases.

Authors: The concentrations and ratios have been given as in previous publications. Thus, to make our publication comparable to those, we would prefer to keep this format for the quantities. However, we have provided the information needed for the calculation of the molar quantities, if needed.

Reviewer #3: Ok.

d. It is not really clear to me which functional groups are involved in the hydrogel formation and

functionalization, especially in the incorporation of growth factors and GAGs, and what the molar ratios of ELR : growth factor : GAG are.

Authors: Thank you for this comment, all the data is now provided for the molar quantities. On line 615-616 it can be seen that HE5-C has 3-4 lysines modified with cyclooctyne and HRGF6-N3 has 12-14 lysines modified with Azide. Each synthetic glycosaminoglycan has a backbone with one azide.

Reviewer #3: Thus, the GAG is not really a building block of the network itself. Maybe, write this a bit clearer?

Is the growth factor added during the production of the hydrogel materials or are the materials loaded with the proteins afterwards? Apparently, this is handled differently for the hydrogels and the cryogels.

Authors: The growth factors for the hydrogel evaluation are added during production but as the liquid is sublimated during freeze-drying for the cryogel production, it is then added afterwards.

Reviewer #3: Thanks for the explanation. Maybe, it should be also added to the manuscript. Thus, for the cryogels the amount of growth factor loaded to the cryogel is not really known and probably different from that incorporated into the hydrogel.

Another remark: The growth factor will not be removed from the scaffold via sublimation. Just the water will be removed.

Please also compare e.g., doi: 10.1016/J.BIOMATERIALS.2021.121170, where growth factor loaded cryogel scaffolds were snap frozen and freeze-dried. The loaded factor was still active and the release of the factor was not changed compared to a freshly loaded cryogel sample.

e. The cross-linking chemistry for the different systems should be please explained more clearly in a scheme. So far, single important information is given in the Results, Discussion and Methods Part and the reader need somehow to combine it.

Authors: The modification and cross-linking chemistry have been now described in the methods section of the article as follows: (Line: 604-612)

The chemical modification of the ELRs was performed following well established methods, as described by Gonzalez de Torre et al.(53) . The aim was to enable a strain-promoted azide alkyne cycloaddition (SPAAC) 'click chemistry' reaction (54) in order to achieve the cross-linking of two different ELRs to form a network that gives a stable scaffold (hydrogel or cryogel). To this end, azide (N3) groups were introduced through the modification of the free ε-amine of the lysine residues present in the HRGD6 ELR, giving HRGD6-N3. Similarly, cyclooctyne groups were conjugated to the HE5, to render a HE5-C. Both modified ELRs were used for the SPAAC reaction and hydrogel/cryogel formation. Table 4 shows the modification degree obtained for the different ELRs.

Further information and a scheme can be found in the seminal article about this cross-linking approach using ELRs (also referenced in the manuscript):

<https://www.sciencedirect.com/science/article/pii/S1742706114000580>

Reviewer #3: Adding this section to the Methods part very much improved the understanding of the crosslinking chemistry for the reader.

7. Characterization of the hydro/cryogel materials:

a. So far the authors present the mechanical characterization of only the pure ELR cryogels with different pore sizes (two methods were used to determine storage modulus and Young's modulus; the values obtained are not compared or discussed further). A comparison with GAG-containing cryogels would have been useful to see if/how the GAG changes the mechanical properties of the materials.

Authors: The added GAGs will bind to approximate 1/500 of the total binding sites available for cross linkage, thus having a minimal impact on the overall cross linkage amount.

Reviewer #3: Probably the GAG does not change the properties very much. Maybe, the authors should include this statement in the text. For many samples E is around 3-times G' . The authors do not explain why they determined both storage modulus and Young's modulus; the values obtained are not compared or discussed further.

b. Also, the pore structure is only shown for the pure ELR cryogels and not for the GAG-containing ones, which are then ultimately used in the in vivo experiment. If it is not known whether the pore structure and the mechanical properties are the same as in the control, it cannot be ruled out that changes in the cell response also result from a changed overall or local stiffness (which is not determined at all) or pore geometry and not only from the release of the growth factor.

Authors: Please see the new supplement figure 2 A and B where A shows the pore size without the added GAGs and B shows pore size with GAGs added demonstrating no observable difference

Reviewer #3: Roughly by eye there are no big difference in the appearance of the dry (!) materials. I think it is ok as it is now.

c. It is a nice result that macroporous hydrogels (cryogels) with different pore sizes can be produced, but the influence of the pore size on the in vivo response is not investigated (cf. also above, influence of the GAG component).

Authors: Thank you for this comment, for more in depth evaluation on how pore size influences the in vivo response we recommend the following articles [10.1016/j.actbio.2013.12.042](https://doi.org/10.1016/j.actbio.2013.12.042) and [10.1038/s41467-019-11397-1](https://doi.org/10.1038/s41467-019-11397-1)

Reviewer #3: Thank you for mentioning the relevant references. However, I meant it would have been nice to show this for the material presented. These experiments are not required but the authors should maybe mention that they plan such experiments and cite the articles given above in this context.

d. For the sake of completeness, the mechanical properties of the ELR and ELR-GAG hydrogels should also be determined and shown.

Authors: We thank the reviewer for the comment, which we have addressed for the ELR hydrogels. The results obtained can be found as Supplementary figures in the new version of the manuscript, and described in the results section of the manuscript, as follows: (Line 255-258, and 652-687)

On the other hand, the ELR hydrogel gave a tangential E value of 4041.6 ± 325.8 Pa (Supplementary Figure 2), while the G' was found to be 633.8 ± 95.9 Pa (see Supplementary Figure 3 for the strain sweep curve). Both measurements resulted in a much higher stiffness than for all the groups of the cryogel counterparts.

The methods and discussion sections have also been updated to include the mechanical measurements of the ELR hydrogels.

As regards GAG-modified hydrogels, they are not expected to show changes in the mechanical properties compared to the unmodified ones, considering the low degree of conjugation (approximately 1 out of 500 ELR molecules modified with the GAG) and previous works where a low modification has not led to significant changes in hydrogel morphology nor mechanical properties (e.g., <https://www.sciencedirect.com/science/article/pii/S174270612100372X> or <https://link.springer.com/article/10.1007/s10856-019-6232-z>).

Reviewer #3: The experimental results were added.

For me the region the curve in the region between 0-30% compression does not look linear. How was the linear regression made in this region?

e. In my opinion, the most important point missing regarding the characterization of the ELR and ELR-GAG hydrogel materials to substantiate the study and make it convincing are studies on the uptake and release of the growth factors in the GAG-modified hydrogels and cryogels. The binding studies on the surface-bound GAGs by means of the screening assay and SPR can certainly only provide initial indications of the binding and release behavior of various growth factors to GAG-containing hydrogel materials, but cannot be used alone to interpret the in vitro and in vivo data shown and cell-instructive properties of the hydrogel materials in general. It is expected that the comparison of uptake and release of VEGF and FGF-2 from GAG nr. 19-containing cryogels will provide important clues to better understand the observed in vivo response.

Authors: This question is of course of great interest, and we have therefore tried to answer this by setting up an experiment where we loaded VEGF into two groups of cryogels, one made with GAGs and one without. We had a lot of problems with too high and too low signals, probably due to VEGF sticking to the plastic in tubes and pipette tips. After some optimization we were able to produce the graph below but unfortunately the cryogels with GAGs showed too high signal (as can be seen with the values being the absorbance maximum of around 4) and cryogels with no GAGs had technical difficulties. The graph indicates a release of VEGF from the cryogels functionalized with GAGs a half time of around 3h. Compared to our SPR data for GAG and VEGF, a half time of around 20 seconds was measured. If this data is representative, the bulk release from the cryogel was around 500 times slower compared the GAG-VEGF complex alone. That the bulk release would be slower is often the case as a VEGF molecule can jump from one GAG to another, thus stays in the cryogel for a longer time.

Reviewer #3: Unfortunately, I do not really understand the experimental set-up. What is the difference between the 2 curves given in the diagram? Was the absorbance of the protein-loaded cryogels/hydrogels measured? Usually, the release of growth factors from hydrogel materials is measured in the supernatant (please compare the relevant literature) using ELISA analysis. The authors should repeat the experiments or at least mention that such experiments need to be done as they could very much help to interpret the in vitro and in vivo results. The developed systems should be defined also regarding growth factor release.

f. Why is the combination of FGF-2 with additional VEGF only investigated in the cryogels/in vivo and not in the hydrogels/in vitro experiments?

Authors: The main aim of our modular system was to show its usability for tissue regeneration in vivo and how it can be used in vitro to control cell activity in a completely defined system. In vitro, the access of cells was a limitation and to show proof-of-concept in this setting we chose just to optimize for FGF2. In vivo, according to literature the formation of blood vessels is known to require both the activation of VEGF and FGF2, which is why we chose this combination.

Reviewer #3: Thanks for the explanation.

As mentioned by the authors according to literature the formation of blood vessels is known to require both the activation of VEGF and FGF2. Nevertheless, in this study they report that VEGF has no additional effect.

The explanation given is based on wrong numbers (compare 5e.). The discussion is still based on the wrong values for the K_d , i.e. equal values for VEGF and FGF-2. However the K_d for VEGF is much higher!

In my opinion the following paragraph need to be corrected accordingly (otherwise results part and discussion are inconsistent), line 444-451:

"However, we saw no added effect of including VEGF in the macroporous ELR, when combining it with synthetic glycosaminoglycans. We suggest that this was due to the 100 times faster release (K_{off}) of VEGF compared to FGF2 for the selected synthetic glycosaminoglycan, despite virtually having the same equilibrium dissociation constant (K_d). This indicates in turn that focus should be on K_{off} and not on K_d when looking to sustain the efficacy of growth factors when using synthetic

glycosaminoglycans.”

g. The addition of VEGF showed no significant effect in the in vivo experiment. However, only 1/5 of the mass of FGF-2 was used to load the cryogels. Furthermore, the loading efficiency may be different for the two factors. However, this was not determined. The analysis of the release behavior of the two different factors from the cryogel materials could be more helpful in the interpretation of the data than the comparison of the K_d values determined on surface-bound GAGs.

Authors: The values used were taken from this very nice article, 10.1242/jcs.02483, as too much growth factors delivered also gives adverse effects.

Reviewer #3: Please compare comments to 7e.)

h. Did the authors check whether autoclavation changes the properties of the cryogels?

Authors: The cryogels did shrink to an extent during autoclaving but similar to the non-autoclaved ones they recover their size when hydrated.

Reviewer #3: Thanks for the explanation.

8. The authors should explain why they use a hydrogel for the in vitro experiments and a macroporous cryogel for the in vivo experiments. Why is the morphogenesis of HUVECs studied in 2D and not in 3D in the hydrogel system (especially since the abstract also mentions 3D in vitro systems)?

Authors: As a proof of principle, to show that synthetic glycosaminoglycans linked to the ELR gel were able to interact with cell surface receptors also after immobilization in the hydrogel, we chose an assay with an easy and clear readout and therefore we chose the HUVEC tube formation assay. This is a 2D assay and the abstract has been updated to reflect this.

Reviewer #3: Ok. However, there are also a lot of 3D HUVEC tube formation assays described in literature, that better reflect the in vivo situation (these should also work as the authors used MMP-cleavable hydrogels) - just as a remark.

9. The technical terms should be used correctly. Firstly, it is confusing when a capital K is used for both the thermodynamic quantity equilibrium dissociation constant (K_d) and the kinetic constants K_{on} and K_{off} . Usually, a small k is used for the kinetic velocity constants. Also, it should correctly read “equilibrium dissociation constant”, “dissociation rate constant” and “association rate constant” instead of “total binding” or “equilibrium constant”, “dissociation constant” and “association constant”.

Accordingly, on page 6, line 118, it should also correctly read “... on a kinetic and thermodynamic level ...”.

Authors: Thank you for pointing this out. It has been corrected for in the text.

Reviewer #3: Ok. Please compare 5e.)

I think the authors do not mean “naïve” but “native” and not “capsel” but “capsule (e.g. Figure SI1, I)”.

Authors: The capsule naming has been updated

The manuscript has been updated and the naming have been changed from “naïve” to “control”

Reviewer #3: Ok. Moreover, Sigma-Aldrich should be replaced by Sigma-Aldrich please.

10. Some statistical data are missing, for instance in Figure 1 A and C the authors do not indicate what is plotted here and what is shown by the error bars. Please add to the Figure Legend or add a

separate section "Statistics", where you give this important information.

Authors: The figure legend has been updated with the following text:

For A and C data is shown in a box and whisker (as Tukey) of 36 binding spots for each synthetic glycosaminoglycans

Similarly, for the determination of K_d , K_{on} and K_{off} , the number of measurements and the errors are not given (neither in the manuscript nor in the tables in the SI). Was only one experiment carried out?

Authors: All growth factors for the SPR experiments were run at 5 different concentrations and the trueness of the fit, χ^2 is available in the supplement for each growth factor.

Reviewer #3: Thanks for adding relevant information.

11. The References should be in a uniform format. Ref. 14 contains two-times the term "Author manuscript". Sometimes doi numbers are given, sometimes not.

Authors: Thank you the manuscript has been updated to use natures referencing style.

Reviewer #3: Please check references 4 and 5.

12. The labels in the figures are in part clearly too small.

Authors: We agree that the labels should be easily readable, and this will be updated when we know the final size for print of each figure.

Reviewer #3: The authors need to check whether they used the correct Figure numbers in the text (and SI) after the modifications please.

New comments from the reviewers are marked in blue while new answers from the authors are marked in purple. The original comments are marked in black.

Reviewers' comments:

Reviewer #1 (Remarks to the Author):

The authors have suitably revised the manuscript, including the addition of new data. I therefore think that the manuscript is suitable for publication.

Reviewer #2 (Remarks to the Author):

The authors have responded to all the comments satisfactorily.

Reviewer #3 (Remarks to the Author):

Reviewers' comments together with answers:

Reviewer #3 (Remarks to the Author):

Reviewer #3: The manuscript has been very much improved. However, in my opinion there are still some points that need to be addressed/considered. Please compare my comments below ("Reviewer #3: ..." inserted after the authors' answers). Figures have been removed.

Manuscript Number: COMMSBIO 22 1236 T

The authors investigated the potential of chemoenzymatically synthesized glycosaminoglycans (GAGs) for controlled binding and release of growth factors. After screening a library of 52 GAGs, selected synthetic GAGs were used to functionalize nanoporous or macroporous hydrogels based on elastin-like recombinamers (ELRs). These materials were loaded with pro-angiogenic growth factors to direct cellular response in a controlled manner. As a proof-of-concept (a) in vitro the tubular morphogenesis of endothelial cells seeded on hydrogels and (b) in vivo the tissue response upon subcutaneous implantation of macroporous hydrogels into a mouse model were studied.

The idea and general concept of incorporating synthetic GAGs into hydrogels to modulate growth factor concentration is interesting and would overcome many problems of hydrogels based on naturally derived GAGs that are highly heterogeneous

[doi.org/10.1021/acs.accounts.9b00420]. However, the authors should nevertheless mention in the manuscript that naturally derived unmodified and modified GAGs have already been successfully used for many years in hydrogels for such purposes - relevant papers should be cited (please also compare Specific comments). It has also been shown for naturally derived GAGs (often heparin) that by varying the degree of sulfation of the GAG and/or the GAG concentration in the hydrogel, uptake and release of signalling proteins can be controlled (e.g., doi.org/10.1016/j.biomaterials.2018.07.056, doi: 10.1002/ADFM.202000068, doi: 10.1039/C9FD00016J). Macroporous GAG-based hydrogels were also frequently used for sustained protein delivery (e.g., doi:

10.1016/J.BIOMATERIALS.2021.121170), for instance to induce vascularization like in the paper under consideration (doi: 10.1016/j.actbio.2014.11.002, doi: 10.1002/mabi.202100234 or doi: 10.1021/acs.biomac.8b00331). This “State of the Art” should be added to the manuscript.

Another major criticism is that the authors do not show whether the tunability of binding and release of growth factors by the synthetic GAGs that they found for surface-bound GAGs is maintained when the GAGs are incorporated into hydrogel networks and whether similar trends are found here as for the surface-bound GAGs. As only a single GAG was selected to produce the hydrogel materials used for the in vitro and in vivo experiments, the advantages and a convincing evidence of the modularity of a platform based on different synthetic GAG-based hydrogels are not really shown experimentally. Regarding the ELR component of the hydrogels the authors mention in the introduction that "Due to the recombinant expression, several modifications can be made, such as introducing a RGD motif for attachment, a matrix metalloproteinase domain for controlled degradation or functionalization of lysine groups for further modification." (page 5, lines 98-100), but later do not further explain the properties of the specific ELR that they used in their study (see below). Here, too, it would have made sense to compare at least two different ELRs to really demonstrate the modularity of the material system.

The Methods part should please be improved.

Some calculations and interpretations are not comprehensible to me or wrong in my opinion (details see below).

Specific comments:

1. The title could have been chosen more appropriately to emphasize that this study also reports about biomaterial development. Moreover, it is not the elastin-like recombinamers that are macroporous, but the hydrogels made from them.

Authors: The title has been updated from:

Synthetic glycosaminoglycans and macroporous elastin-like recombinamers as a modular platform for tissue regeneration

To: (Line 1-2)

Controlled release of growth factors using synthetic glycosaminoglycans in a modular macroporous scaffold for tissue regeneration

Reviewer #3: The updated title fits better. However, a suggestion from my side would be the following title: Macroporous scaffolds based on synthetic glycosaminoglycans and elastin-like recombinamers as a modular platform for tissue regeneration

Authors: Thank you for the suggestion, but after your first comment we carefully thought about a new title to better define our work. Therefore, we prefer to keep the modified title, which summarizes our study more accurately.

2. The selection of the abbreviations listed and explained seems arbitrary. Why, for example, is only the abbreviation of some growth factors explained?

Authors: The abbreviations list has been updated to be more extensive. (Line 23-75)

Reviewer #3: ok

3. The wording of the abstract could be more precise. In particular, I would also mention that K_d values were determined.

I suggest naming the technique used to introduce the macropores not as cryogelation; it is rather a combination of porogen leaching and cryogelation.

In the abstract the authors do not mention that they also tested VEGF loaded macroporous hydrogels for the in vivo studies.

Authors: The abstract has been updated to include these suggestions but there was also a need to shorten it to meet the format guidelines of approximate 150 words: (Line 77-90)

Healthy regeneration of tissue relies on a well-orchestrated release of growth factors. Herein, we show the use of synthetic glycosaminoglycans for controlled binding and release of growth factors to induce a desired cellular response. First, we screened glycosaminoglycans with growth factors of interest to define K_{on} (association rate constant), K_{off} (dissociation rate constant), and K_d (equilibrium rate constant). As proof-of-concept, we functionalized an elastin-like recombinamer (ELR) hydrogel with a synthetic glycosaminoglycan and immobilized fibroblast growth factor 2 (FGF2) and/or vascular endothelial growth factor (VEGF), demonstrating that cultured human umbilical vein endothelial cells differentiated into tube-like structures. Taking this concept further, we developed a tunable macroporous ELR cryogel, containing the synthetic glycosaminoglycan and FGF2 that showed increased blood vessel formation and reduced immune response compared to control when implanted in a subcutaneous mouse model. These results demonstrated the possibility for specific release of desired growth factors in a modular structure in vitro and in vivo.

Reviewer #3: I suggest to additionally replace:

... First, we screened glycosaminoglycans with growth factors of interest to define K_{on} (association rate constant), K_{off} (dissociation rate constant), and K_d (equilibrium rate constant). As proof-of-concept, we functionalized an elastin-like recombinamer (ELR) hydrogel with a synthetic glycosaminoglycan and immobilized fibroblast growth factor 2 (FGF2) and/or vascular endothelial growth factor (VEGF), demonstrating that cultured human umbilical vein endothelial cells differentiated into tube-like structures. Taking this concept further, we developed a tunable macroporous ELR cryogel, containing the synthetic glycosaminoglycan and FGF2 that showed increased blood vessel formation and reduced immune response compared to control when implanted in a subcutaneous mouse model. These results demonstrated the possibility for specific release of desired growth factors in a modular structure in vitro and in vivo.

By:

... First, we screened glycosaminoglycans with growth factors of interest to determine k_{on} (association rate constant), k_{off} (dissociation rate constant), and K_d (equilibrium rate constant). As proof-of-concept, we functionalized an elastin-like recombinamer (ELR) hydrogel with a synthetic glycosaminoglycan and immobilized fibroblast growth factor 2 (FGF2), demonstrating that cultured human umbilical vein endothelial cells on top differentiated into tube-like structures. Taking this concept further, we developed a tunable macroporous ELR cryogel material, containing a synthetic glycosaminoglycan and FGF2 that showed increased blood vessel formation and reduced immune response compared to control when implanted in a subcutaneous mouse model. These results demonstrated the possibility for specific release of desired growth factors in/from a modular 3D scaffold in vitro and in vivo.

(As far as I understood VEGF was not used in the in vitro studies.)

Authors: Thanks for the updated abstract, line 82-92 have been updated to reflect these changes

4. The introduction could be more conclusive and include more references to the current "State of the Art". The authors mention only few examples of how in general sustained release from biomaterials can be achieved. More strategies reported in original articles and review articles could be added here.

For instance, the review article doi: 10.1039/C3TB20853B summarizes "Chemical strategies for the presentation and delivery of growth factors". The strategy pursued in nature of adjusting growth factor gradients via GAGs, which the authors describe as very advantageous, has also been used in the design of biomaterials to modulate growth factor concentrations and is in principle not new (doi:10.3389/fcell.2021.760532).

The authors should mention this and cite some relevant previous studies, e.g. doi: 10.1016/j.actbio.2014.11.002, doi: 10.1002/mabi.202100234, doi: 10.1002/ADFM.202000068, doi: 10.1021/acs.biomac.8b00331 or doi: 10.1016/J.BIOMATERIALS.2021.121170, to name only some examples of heparin-based hydrogels.

From the introduction in its present form the exact intention of the study is not clear. It is argued somewhat arbitrarily with various aspects of ECM, which are apparently to be imitated by the materials presented here. Why both nanoporous and macroporous systems were investigated is not plausible for me. In part, the impression is given that macroporous hydrogels are generally better suited for tissue regeneration and that their too fast release of growth factors is a problem that should be solved using principles of nature, i.e. by creating GAG-based systems. A specific comparison of the introduced materials with and a differentiation from already established GAG-based hydrogel systems is missing.

Some explanations are not coherent from my point of view: Page 4, line 70/71: „... by having growth factors incorporated into scaffolds such as hydrogels, where the release is limited due to the low area to volume ratio.”

Authors: Thank you, the suggested articles are indeed highly relevant and have now been added to the introduction. (Line 112-124)

Reviewer #3: The authors added some relevant references to the introduction. However, some aspects already mentioned in the general part of my first review were not considered in this context. In particular, in my opinion the following text parts have to be revised: (line 108 -122) "The disadvantage of manufactured macroporous scaffolds incorporated with growth factors is quick release due to high exposure area. To circumvent the fast release from macroporous structures, different strategies have been designed¹¹; using aptamers¹², or creating a binding tag ^{12,13} . However, these approaches only allow for either binding of one single factor or only the loaded factors and not binding factors produced in the body. ... This, in combination with the lack of sequencing technologies, have led to the use of long undefined glycosaminoglycans purified from different origins^{16–20} , that bind growth factors but lack the possibility of fine tuning release properties ²¹ ."

As already mentioned in the general first introducing paragraph of my original comments, it has also been shown for hydrogels based on naturally derived GAGs (often heparin) that by varying the degree of sulfation of the GAG and/or the GAG concentration in the hydrogel, uptake and release of signalling proteins can be controlled (e.g., doi.org/10.1016/j.biomaterials.2018.07.056, doi: 10.1002/ADFM.202000068, doi: 10.1039/C9FD00016J). The same basic concept also enabled the development of macroporous GAG-based hydrogels for sustained and fine-tuned protein delivery (e.g., doi: 10.1016/J.BIOMATERIALS.2021.121170) (thus, these materials do not "lack the possibility of fine tuning release properties" or have the disadvantage of "quick release due to high exposure area" as stated by the authors).

Authors:

Thank you for this comment, line 150-152 have been updated from "This, in combination with the lack of sequencing technologies, have led to the use of long undefined glycosaminoglycans purified from different origins, that bind growth factors lack the possibility of fine tuning release properties "

To

"This, in combination with the lack of sequencing technologies, have led to the use of long undefined glycosaminoglycans purified from different origins that bind growth factors, thus with unknown sugar length and sulfation sequence."

5. Characterization of GAG-growth factor interactions:

a. Although precise information on the sequences of the various synthetic GAGs can be found in the SI, it would be helpful for the reader to mention some information on the GAGs that could play a role in growth factor binding directly in the manuscript, e.g. as a table summarizing the degree of sulfation, the molar mass, the number of negatively charged groups (e.g. also carboxyl groups that could mediate the binding of growth factors to non-sulfated GAGs, such as GAG no. 3, see p. 6, line 130/13). So far, only the numbers assigned

to the GAGs are given here, or the sequences of the selected GAGs are shown very small in the axis in Figure 1. In addition, the specification of important properties of the growth factors used, such as molar mass, IEP, dimensions, would help to better interpret/understand the interaction with the GAGs. The analysis should therefore be based more on physicochemical properties of the components and be comprehensible to the reader through a systematic presentation. In particular, a comparison with the only natural GAG heparin (no. 1) would perhaps also be interesting here (or at least a hint that heparin is hidden behind no. 1), whereby knowledge of the molar mass (distribution) used would also be interesting here and a comparison with literature data (e.g. with doi:10.3390/molecules24183360).

Authors: For easier understanding of the data, a table for the GAGs (Table 1) and the growth factors (Table 2) have been added to the methods section with the data mentioned above. The data is also compared to the mentioned literature: 10.3390/molecules24183360.

Reviewer #3: The added tables are very helpful for the reader. Unfortunately, the units in Table 1 are missing and should be added by the authors. Moreover, they should explain, what they exactly mean by “number of sulphation’s” and “sugar length”.

Authors: The table has been updated with units and the following text on line 675-678 has been added “The number of monosaccharides, degree of sulfation and number of carboxyl groups were counted for each glycosaminoglycan oligosaccharide”

b. On page 6, lines 114-117, the authors state "The growth factors added to the microarray revealed three groups of glycosaminoglycans: those that showed high binding to all growth factors, those that showed low/no binding and finally those that showed unique binding depending on the growth factor. For group 2, information is given in the legend to Figure 1. For the other two groups, corresponding information should be please given as well. One possibility could be to include it in the additional table on the GAG characteristics.

Authors: To clarify, the text has been changed as shown below and a supplement figure has been added for the microarray screening of additional growth factors

From:

Fig. 1 A-C show vascular endothelial growth factor (VEGF) and hepatocyte growth factor (HGF) as an example of the results obtained with this method. The growth factors added to the microarray revealed three groups of glycosaminoglycans: those that showed high binding to all growth factors, those that showed low/no binding and finally those that showed unique binding depending on the growth factor.

To further study this on a kinetic level, we decided to evaluate a subset of the glycosaminoglycans (one low/no binding, GAG nr 3, one high binding, GAG nr 19, and four with unique binding, GAG nr 10, 26, 34 and 43) using surface plasmon resonance (SPR) (see Supplement Table 2).

To: (Line 151-164)

Fig. 1 A-C show vascular endothelial growth factor (VEGF) and hepatocyte growth factor (HGF), furthermore, BMP4, CXCL12, IL-6, FGF2, PDGF-AA, KGF and TGF beta 1 were also tested and data can be found in Supplement Figure 1. The growth factors added to the microarray revealed three groups of glycosaminoglycans: those that showed high binding to all growth factors, those that showed low/no binding and finally those that showed binding depending on the growth factor.

To further study this on a kinetic level, we decided to evaluate a subset of the glycosaminoglycans using surface plasmon resonance (SPR). One glycosaminoglycan with low to no binding to any growth factors, GAG nr 3, one high binding (except for TGF beta 1), GAG nr 19, and four with binding to only a subset of the growth factors, GAG nr 10, 26, 34 and 43 (see Supplement Table 2).

Reviewer #3: The additional information and changes very much improved this part of the manuscript. : I recommend to replace:

“... those that showed high binding to all growth factors, those that showed low/no binding and finally those that showed binding depending on the growth factor.”

By:

“... those that showed high binding to all growth factors, those that showed low/no binding and finally those that showed selective binding depending on the growth factor.”

Author: Thank you for this comment, line 197-199 have been updated with this suggestion

c. In addition, there is a certain discrepancy between the number of GAGs examined in the text (52), in the table (nr. 1-53=54) and in Figure 1 (nr. 0 - 53=54). A clear allocation of the GAGs in the figures is therefore not possible.

Authors: For a clearer and easier understanding, the text has been updated from:

To demonstrate the possibility to use synthetic glycosaminoglycans as a tool for customizing the release rate of growth factors, a range of them were screened on a microarray functionalized with 52 different synthetic glycosaminoglycans (see Supplement Table 1).

To: (Line 147-151)

To demonstrate the possibility to use synthetic glycosaminoglycans as a tool for customizing the release rate of growth factors, a range of them were screened on a microarray functionalized with 52 different synthetic glycosaminoglycans, referred to as 2-53 while 1 is commercially available heparin and, when applicable, 0 is background values (see Supplement Table 1).

Reviewer #3: The text is much clearer now. I suggest to additionally replace:

... and, when applicable, 0 is background values (see Supplement Table 1).

By:

... and, when applicable, the background value (0) is given (see Supplement Table 1).

Author: Thanks for this suggestion, line 192-193 have been updated with this suggestion

d. Is the use of equal mass concentrations of the different growth factors useful in microarray screening, as is the use of equal mass concentrations of the antibodies? Would not an equal molar concentration of substances make more sense?

Authors: As there is no way of knowing the number of different binding sites of a glycosaminoglycan and if they are competing for the same binding site as the antibody, the microarray was used as a way to screen factors and thus a very high concentration was used. For a deeper understanding of the binding properties, the glycosaminoglycans were further evaluated using SPR.

What protein concentrations were used for SPR? I could not find any information on this in the manuscript.

Authors: The concentrations used have been added to the methods section and can be seen in Table 2

Reviewer #3: ok

e. The formula given on page 25 is not correct in my opinion. It should correctly read $K_d = K_{off}/K_{on}$.

Authors: Thank you for pinpointing this issue. We have corrected it in the revised version of the manuscript.

Reviewer #3: Ok, however, it is common to use a small k (and not a capital K) for the kinetic velocity constants (thus: kon and koff). The authors please should change it accordingly.

Authors: Thank you for pointing this out. It has been updated accordingly.

Furthermore, the authors should definitely check all the values for K_d given in the SI (12692_0_data_set_98338_r3frs0.xlsx). I calculated different values except for the two growth factors CXCL12 and CXCL4. This would also change the presentation in the Results or Discussion Part in some cases. For example, for GAG nr. 19 I do not calculate the same K_d for VEGF (410 nM) and (FGF-2 (413 nM), as the authors state and interpret on page 7, line 137 and page 20/21, lines 397, but a 3 times higher K_d for VEGF (136 nM) than for FGF-2 (51 nM) underpinning the higher effect for FGF-2 found in the cell experiment. It is questionable whether the statement that "focus should be on release rate and not total binding when looking to sustain the efficacy of growth factors when using synthetic glycosaminoglycans" can still be sufficiently substantiated.

Authors: Thank you very much for going into the supplement and finding this mistake. Due to an Excel error, the K_d values were not correctly calculated from K_{on} and K_{off} , Figure 1 has been updated and shows the corrected values, as can be seen below, as well as in supplement table 2.

Reviewer #3: The mentioned error was corrected in the results part. However, the discussion is still based on the wrong values for the K_d , i.e. equal values for VEGF and FGF-2.

The Kd for VEGF is much higher! It is questionable whether the statement that "focus should be on release rate and not total binding when looking to sustain the efficacy of growth factors when using synthetic glycosaminoglycans" can still be sufficiently substantiated. In my opinion the following paragraph need to be corrected accordingly (otherwise results part and discussion are inconsistent), lines 444-451:

"However, we saw no added effect of including VEGF in the macroporous ELR, when combining it with synthetic glycosaminoglycans. We suggest that this was due to the 100 times faster release (Koff) of VEGF compared to FGF2 for the selected synthetic glycosaminoglycan, despite virtually having the same equilibrium dissociation constant (Kd). This indicates in turn that focus should be on Koff and not on Kd when looking to sustain the efficacy of growth factors when using synthetic glycosaminoglycans."

From:

..

To:

..

Author: Thank you for this valuable comment as it is important to be consistent for easier understanding. The discussion has been updated with the following text on line 539-545

From:

"We suggest that this was due to the 100 times faster release (koff) of VEGF compared to FGF2 for the selected synthetic glycosaminoglycan, despite virtually having the same equilibrium dissociation constant (Kd). This indicates in turn that focus should be on koff and not on Kd when looking to sustain the efficacy of growth factors when using synthetic glycosaminoglycans. But as none of the other synthetic glycosaminoglycans characterized by SPR had a slower koff value for VEGF, further screening is needed."

To:

"We suggest that this was due to the 100 times faster release (koff) of VEGF compared to FGF2 for the selected synthetic glycosaminoglycan, but could also be due to the 2.5 times higher equilibrium dissociation constant (Kd). We believe this indicates that focus should be on koff and not on Kd when looking to sustain the efficacy of growth factors when using synthetic glycosaminoglycans. As none of the synthetic glycosaminoglycans had the exact same Kd and had a large difference in koff, further screening is needed to determine the importance of Kd and koff."

f. Due to the incorrect calculation of the Kd values, the factor 2000 on page 18, line 340 ff, is incorrect in my opinion, too. The correct value would be about 1000.

Authors: This value has been updated in the article.

Reviewer #3: ok

g. The claim made on page 6, lines 124-127 "The synthetic glycosaminoglycan that had the longest carbohydrate chain as well as the highest degree of sulfation, GAG nr 19 (highlighted in orange in Fig. 1E), showed unique binding to transforming growth factor beta 1 (TGF beta 1), keratinocyte growth factor (KGF) and connective tissue growth factor (CCN2)". I cannot understand this on the basis of Figure 1. Is it more likely that the following is meant: "The synthetic glycosaminoglycan with the longest carbohydrate chain as well as the highest degree of sulphation, GAG No. 19 (highlighted in orange in Fig. 1E), was the only one to show binding to transforming growth factor beta 1 (TGF beta 1), keratinocyte growth factor (KGF) and connective tissue growth factor (CCN2)"?

Authors: Thank you. The text has been updated with your suggestion for an easier understanding.

Reviewer #3: ok

6. Production of the hydro-/cryogel materials:

a. The description of hydro- and cryogel production should be optimized. In the current way it is not understandable and not comprehensible.

Authors: The method part has been optimized. For example this section:

Water was sprayed using on top of liquid nitrogen and sieved through strainers with mesh sizes 500, 200, and 100 to collect ice crystals of desired size. Ice crystals were added to a mold together with the dissolved ELR and left in a cryostat at -5 °C to equilibrate.

Have been changed to: (Line: 624 -629)

MilliQ water (Millipore Reference, Merck) was sprayed using a spray bottle on top of liquid nitrogen and sieved through strainers with mesh sizes 500, 200, and 100 µm (Pluristrainer, 43-50500-03, 43-50200-03, 43-50100-51, pluriSelect Life Science) to collect ice crystals of desired sizes. Ice crystals were added to a mold (here a syringe was used (1ml syringe, 329654, BD Bioscience)). Ice crystal and dissolved ELR were left in a cryostat (HM500M, Microm) at -5° C to equilibrate.

Reviewer #3: The description is more detailed and comprehensible now.

b. It should be explained very briefly how the ELRs are produced, even if this work has already been published. I would recommend including a brief description of the properties (e.g., molecular weight, RGD sequences, functionalisation with cyclooctyne and azide).

Authors: This has been described in the methods section, and a table with the ELR sequences and molecular weights has been included (Table 3). The following text has been added to the manuscript: (Line:586-603)

The design of the ELRs used in this work, termed HE5 and HRGD6, has been described elsewhere (26), and table 3 includes their sequences and molecular weights. Each one comprises different bioactive domains. HE5 contains matrix metalloproteinase (MMP)-

sensitive domains that allow biodegradation mainly by MMP-2, MMP-9 and MMP-13 (26) On the other hand, HRGD6 include Arg-Gly-Asp (RGD) for cell attachment and interaction, improving the biocompatibility of the scaffolds made with it, like the macroporous cryogels (51).

Both ELRs were produced as previously described(52) . In brief, the ELR genes were cloned into a pET-25b(+) plasmid (Novagen, Merck) for expression in Escherichia coli (BLR(DE3) strain, Novagen, Merck) in a 15-L bioreactor (Applikon Biotechnology). The ELRs were subsequently purified through consecutive heating and cooling steps, dialyzed against ultra-pure water and filtered through a 0.22 µm filters (Nalgene) prior to freeze-drying.

Reviewer #3: Ok.

c. Instead of using mass concentrations and ratios, it would be probably better to work with molar quantities in some cases.

Authors: The concentrations and ratios have been given as in previous publications. Thus, to make our publication comparable to those, we would prefer to keep this format for the quantities. However, we have provided the information needed for the calculation of the molar quantities, if needed.

Reviewer #3: Ok.

d. It is not really clear to me which functional groups are involved in the hydrogel formation and functionalization, especially in the incorporation of growth factors and GAGs, and what the molar ratios of ELR : growth factor : GAG are.

Authors: Thank you for this comment, all the data is now provided for the molar quantities. On line 615-616 it can be seen that HE5-C has 3-4 lysines modified with cyclooctyne and HRGF6-N3 has 12-14 lysines modified with Azide. Each synthetic glycosaminoglycan has a backbone with one azide.

Reviewer #3: Thus, the GAG is not really a building block of the network itself. Maybe, write this a bit clearer?

Author: Thank you for the comment. On line 675-676 he following sentence is added: "The glycosaminoglycans that were chosen can be seen in Table 1, where each glycosaminoglycan has a starting sugar with an azide."

Is the growth factor added during the production of the hydrogel materials or are the materials loaded with the proteins afterwards? Apparently, this is handled differently for the hydrogels and the cryogels.

Authors: The growth factors for the hydrogel evaluation are added during production but as the liquid is sublimated during freeze-drying for the cryogel production, it is then added afterwards.

Reviewer #3: Thanks for the explanation. Maybe, it should be also added to the manuscript.

Thus, for the cryogels the amount of growth factor loaded to the cryogel is not really known and probably different from that incorporated into the hydrogel.

Thank you for the comment, it is true that there probably is a difference in loading efficacy.

In the methods part it can be found that the hydrogels are made as described on line 834-842:

“ELR was dissolved in PBS at a concentration of 64.28 mg/ml for ELR with cyclooctyne and 35.71 mg/ml for ELR with azide and left at 4°C overnight. Glycosaminoglycan nr 19 was added at a concentration of 2 µg/ml (for a final concentration of 1 µg/ml of ELR mixture) of ELR with cyclooctyne and left at 4°C for 1 hour. The same procedure was performed for FGF2 added at a concentration of 200 ng/ml (final concentration 100 ng/ml) and left at 4°C for 1 hour. Then the gels were mixed with equal parts ELR azide and cyclooctyne, quickly mixed by pipetting using cold pipets and 10 µl mixture was added to an angiogenesis slide (81507, Ibbidi), incubated in at 4°C for 15 min followed by 10 min at 37°C”

While the cryogels are made as described on line 869-872:

“After freeze drying, the cryogels were autoclaved and resuspended in 200 µl of either pure PBS, 200 ng/ml VEGF, 1 µg/ml of FGF2 or both 200 ng/ml VEGF and 1 µg/ml FGF2. Cryogels were left at 4°C for two days and washed with PBS before implantation.”

Another remark: The growth factor will not be removed from the scaffold via sublimation. Just the water will be removed.

Please also compare e.g., doi: 10.1016/J.BIOMATERIALS.2021.121170, where growth factor loaded cryogel scaffolds were snap frozen and freeze-dried. The loaded factor was still active and the release of the factor was not changed compared to a freshly loaded cryogel sample.

e. The cross-linking chemistry for the different systems should be please explained more clearly in a scheme. So far, single important information is given in the Results, Discussion and Methods Part and the reader need somehow to combine it.

Authors: The modification and cross-linking chemistry have been now described in the methods section of the article as follows: (Line: 604-612)

The chemical modification of the ELRs was performed following well established methods, as described by Gonzalez de Torre et al.(53) . The aim was to enable a strain-promoted azide alkyne cycloaddition (SPAAC) ‘click chemistry’ reaction (54) in order to achieve the cross-linking of two different ELRs to form a network that gives a stable scaffold (hydrogel or cryogel). To this end, azide (N₃) groups were introduced through the modification of the free ε-amine of the lysine residues present in the HRGD6 ELR, giving HRGD6-N₃. Similarly, cyclooctyne groups were conjugated to the HE5, to render a HE5-C. Both modified ELRs

were used for the SPAAC reaction and hydrogel/cryogel formation. Table 4 shows the modification degree obtained for the different ELRs.

Further information and a scheme can be found in the seminal article about this cross-linking approach using ELRs (also referenced in the manuscript):

<https://www.sciencedirect.com/science/article/pii/S1742706114000580>

Reviewer #3: Adding this section to the Methods part very much improved the understanding of the crosslinking chemistry for the reader.

7. Characterization of the hydro/cryogel materials:

a. So far the authors present the mechanical characterization of only the pure ELR cryogels with different pore sizes (two methods were used to determine storage modulus and Young's modulus; the values obtained are not compared or discussed further). A comparison with GAG-containing cryogels would have been useful to see if/how the GAG changes the mechanical properties of the materials.

Authors: The added GAGs will bind to approximate 1/500 of the total binding sites available for cross linkage, thus having a minimal impact on the overall cross linkage amount.

Reviewer #3: Probably the GAG does not change the properties very much. Maybe, the authors should include this statement in the text. For many samples E is around 3-times G'. The authors do not explain why they determined both storage modulus and Young's modulus; the values obtained are not compared or discussed further.

Author: We thank the reviewer for the suggestion. We have added the following text to the discussion (lines 514-517):

"Given that the GAGs used in the bioconjugation will bind approximately 1/500 of the total binding sites available for the crosslinking, their presence will have a residual effect on the hydrogel formation process and, therefore, on its structure and mechanical properties".

Regarding the E and the G', both uniaxial compression tests and shear stress rheology were used for the calculation of those parameters, respectively. Considering that some works determine one or the other, we aimed at providing as much data as possible to make our study comparable to others. However, because of the differences in both assays, the values obtained are not directly comparable between them, and they are expected to give differences. Nevertheless, we have included the following sentence to the discussion section (lines 517-522):

"E and G' values obtained by uniaxial compression tests and shear rheology, respectively, give slightly different results due to the inherent differences of the two methods used to characterize the mechanical properties of the hydro and cryogels. While we could have chosen to use only one of them, we aimed at giving as much data as possible to allow the comparison with other studies, where the authors show either E or G' for their constructs".

b. Also, the pore structure is only shown for the pure ELR cryogels and not for the GAG-containing ones, which are then ultimately used in the in vivo experiment.

If it is not known whether the pore structure and the mechanical properties are the same as in the control, it cannot be ruled out that changes in the cell response also result from a changed overall or local stiffness (which is not determined at all) or pore geometry and not only from the release of the growth factor.

Authors: Please see the new supplement figure 2 A and B where A shows the pore size without the added GAGs and B shows pore size with GAGs added demonstrating no observable difference

Reviewer #3: Roughly by eye there are no big difference in the appearance of the dry (!) materials. I think it is ok as it is now.

c. It is a nice result that macroporous hydrogels (cryogels) with different pore sizes can be produced, but the influence of the pore size on the in vivo response is not investigated (cf. also above, influence of the GAG component).

Authors: Thank you for this comment, for more in depth evaluation on how pore size influences the in vivo response we recommend the following articles [10.1016/j.actbio.2013.12.042](https://doi.org/10.1016/j.actbio.2013.12.042) and [10.1038/s41467-019-11397-1](https://doi.org/10.1038/s41467-019-11397-1)

Reviewer #3: Thank you for mentioning the relevant references. However, I meant it would have been nice to show this for the material presented. These experiments are not required but the authors should maybe mention that they plan such experiments and cite the articles given above in this context.

Author: Thank you for the suggestion, the following text has been added to the discussion line 581-583:

“For the in vivo studies only one pore size was tested (200-500 μm), so further studies should be performed to evaluate the other sizes as larger pores have shown to be more anti-inflammatory”

d. For the sake of completeness, the mechanical properties of the ELR and ELR-GAG hydrogels should also be determined and shown.

Authors: We thank the reviewer for the comment, which we have addressed for the ELR hydrogels. The results obtained can be found as Supplementary figures in the new version of the manuscript, and described in the results section of the manuscript, as follows: (Line 255-258, and 652-687)

On the other hand, the ELR hydrogel gave a tangential E value of 4041.6 ± 325.8 Pa (Supplementary Figure 2), while the G' was found to be 633.8 ± 95.9 Pa (see Supplementary Figure 3 for the strain sweep curve). Both measurements resulted in a much higher stiffness than for all the groups of the cryogel counterparts.

The methods and discussion sections have also been updated to include the mechanical measurements of the ELR hydrogels.

As regards GAG-modified hydrogels, they are not expected to show changes in the mechanical properties compared to the unmodified ones, considering the low degree of conjugation (approximately 1 out of 500 ELR molecules modified with the GAG) and previous works where a low modification has not led to significant changes in hydrogel morphology nor mechanical properties (e.g., <https://www.sciencedirect.com/science/article/pii/S174270612100372X> or <https://link.springer.com/article/10.1007/s10856-019-6232-z>).

Reviewer #3: The experimental results were added.

For me the region the curve in the region between 0-30% compression does not look linear. How was the linear regression made in this region?

Author: The reviewer is right and the 0-30% region is not as linear as others in the graph ($R^2 = 0.84$ for a linear regression). Therefore, we have revised our data and decided to use the 10-20% strain region to calculate E ($R^2 = 0.99$) for the hydrogel. In this case, $E = 3077.4 \pm 98.6$ Pa. This information has been updated in the manuscript.

e. In my opinion, the most important point missing regarding the characterization of the ELR and ELR-GAG hydrogel materials to substantiate the study and make it convincing are studies on the uptake and release of the growth factors in the GAG-modified hydrogels and cryogels. The binding studies on the surface-bound GAGs by means of the screening assay and SPR can certainly only provide initial indications of the binding and release behavior of various growth factors to GAG-containing hydrogel materials, but cannot be used alone to interpret the in vitro and in vivo data shown and cell-instructive properties of the hydrogel materials in general. It is expected that the comparison of uptake and release of VEGF and FGF-2 from GAG nr. 19-containing cryogels will provide important clues to better understand the observed in vivo response.

Authors: This question is of course of great interest, and we have therefore tried to answer this by setting up an experiment where we loaded VEGF into two groups of cryogels, one made with GAGs and one without. We had a lot of problems with too high and too low signals, probably due to VEGF sticking to the plastic in tubes and pipette tips. After some optimization we were able to produce the graph below but unfortunately the cryogels with GAGs showed too high signal (as can be seen with the values being the absorbance maximum of around 4) and cryogels with no GAGs had technical difficulties. The graph indicates a release of VEGF from the cryogels functionalized with GAGs a half time of around 3h. Compared to our SPR data for GAG and VEGF, a half time of around 20 seconds was measured. If this data is representative, the bulk release from the cryogel was around 500 times slower compared the GAG-VEGF complex alone. That the bulk release would be slower is often the case as a VEGF molecule can jump from one GAG to another, thus stays in the cryogel for a longer time.

Reviewer #3: Unfortunately, I do not really understand the experimental set-up. What is the difference between the 2 curves given in the diagram? Was the absorbance of the protein-loaded cryogels/hydrogels measured? Usually, the release of growth factors from hydrogel materials is measured in the supernatant (please compare the relevant literature) using ELISA analysis.

The authors should repeat the experiments or at least mention that such experiments need to be done as they could very much help to interpret the in vitro and in vivo results. The developed systems should be defined also regarding growth factor release.

Author: thank you for this valuable comment, the following has been added to the discussion to highlight the need for more deeply understanding the bulk release (see line 584-588):

“Furthermore, the bulk release of growth factors from the cryogels should be evaluated as growth factors may jump from one glycosaminoglycan to another, which may result in a longer retention time in the cryogel. Additionally, binding of growth factors should be tested in regards to the volume of cryogels as well as the amount of glycosaminoglycans in order to see how these factors contribute.”

f. Why is the combination of FGF-2 with additional VEGF only investigated in the cryogels/in vivo and not in the hydrogels/in vitro experiments?

Authors: The main aim of our modular system was to show its usability for tissue regeneration in vivo and how it can be used in vitro to control cell activity in a completely defined system. In vitro, the access of cells was a limitation and to show proof-of-concept in this setting we chose just to optimize for FGF2. In vivo, according to literature the formation of blood vessels is known to require both the activation of VEGF and FGF2, which is why we chose this combination.

Reviewer #3: Thanks for the explanation.

As mentioned by the authors according to literature the formation of blood vessels is known to require both the activation of VEGF and FGF2. Nevertheless, in this study they report that VEGF has no additional effect.

The explanation given is based on wrong numbers (compare 5e.). The discussion is still based on the wrong values for the K_d , i.e. equal values for VEGF and FGF-2. However the K_d for VEGF is much higher!

In my opinion the following paragraph need to be corrected accordingly (otherwise results part and discussion are inconsistent), line 444-451:

“However, we saw no added effect of including VEGF in the macroporous ELR, when combining it with synthetic glycosaminoglycans. We suggest that this was due to the 100 times faster release (K_{off}) of VEGF compared to FGF2 for the selected synthetic glycosaminoglycan, despite virtually having the same equilibrium dissociation constant (K_d). This indicates in turn that focus should be on K_{off} and not on K_d when looking to sustain the efficacy of growth factors when using synthetic glycosaminoglycans.”

g. The addition of VEGF showed no significant effect in the in vivo experiment. However, only 1/5 of the mass of FGF-2 was used to load the cryogels. Furthermore, the loading efficiency may be different for the two factors. However, this was not determined. The analysis of the release behavior of the two different factors from the cryogel materials could be more helpful in the interpretation of the data than the comparison of the K_d values determined on surface-bound GAGs.

Authors: The values used were taken from this very nice article, 10.1242/jcs.02483, as too much growth factors delivered also gives adverse effects.

Reviewer #3: Please compare comments to 7e.)

Author: Please see comment 7e.

h. Did the authors check whether autoclavation changes the properties of the cryogels?

Authors: The cryogels did shrink to an extent during autoclaving but similar to the non-autoclaved ones they recover their size when hydrated.

Reviewer #3: Thanks for the explanation.

8. The authors should explain why they use a hydrogel for the in vitro experiments and a macroporous cryogel for the in vivo experiments. Why is the morphogenesis of HUVECs studied in 2D and not in 3D in the hydrogel system (especially since the abstract also mentions 3D in vitro systems)?

Authors: As a proof of principle, to show that synthetic glycosaminoglycans linked to the ELR gel were able to interact with cell surface receptors also after immobilization in the hydrogel, we chose an assay with an easy and clear readout and therefore we chose the HUVEC tube formation assay. This is a 2D assay and the abstract has been updated to reflect this.

Reviewer #3: Ok. However, there are also a lot of 3D HUVEC tube formation assays described in literature, that better reflect the in vivo situation (these should also work as the authors used MMP-cleavable hydrogels) - just as a remark.

Arthur: Thank you for this comment.

9. The technical terms should be used correctly. Firstly, it is confusing when a capital K is used for both the thermodynamic quantity equilibrium dissociation constant (Kd) and the kinetic constants Kon and Koff. Usually, a small k is used for the kinetic velocity constants. Also, it should correctly read "equilibrium dissociation constant", "dissociation rate constant" and "association rate constant" instead of "total binding" or "equilibrium constant", "dissociation constant" and "association constant". Accordingly, on page 6, line 118, it should also correctly read " ... on a kinetic and thermodynamic level ...".

Authors: Thank you for pointing this out. It has been corrected for in the text.

Reviewer #3: Ok. Please compare 5e.)

Author: 5e has been updated.

I think the authors do not mean “naïve” but “native” and not “capsel” but “capsule (e.g. Figure S11, I)”.

Authors: The capsule naming has been updated

The manuscript has been updated and the naming have been changed from “naïve” to “control”

Reviewer #3: Ok. Moreover, Sigma-Aldrish should be replaced by Sigma-Aldrich please.

Author: Thank you, this has now been updated.

10. Some statistical data are missing, for instance in Figure 1 A and C the authors do not indicate what is plotted here and what is shown by the error bars. Please add to the Figure Legend or add a separate section “Statistics”, where you give this important information.

Authors: The figure legend has been updated with the following text:

For A and C data is shown in a box and whisker (as Tukey) of 36 binding spots for each synthetic glycosaminoglycans

Similarly, for the determination of K_d , K_{on} and K_{off} , the number of measurements and the errors are not given (neither in the manuscript nor in the tables in the SI). Was only one experiment carried out?

Authors: All growth factors for the SPR experiments were run at 5 different concentrations and the trueness of the fit, χ^2 is available in the supplement for each growth factor.

Reviewer #3: Thanks for adding relevant information.

11. The References should be in a uniform format. Ref. 14 contains two-times the term “Author manuscript”. Sometimes doi nummers are given, sometimes not.

Authors: Thank you the manuscript has been updated to use natures referencing style.

Reviewer #3: Please check references 4 and 5.

Author: References 4 and 5 have been updated to better follow Nature’s referencing style.

12. The labels in the figures are in part clearly too small.

Authors: We agree that the labels should be easily readable, and this will be updated when we know the final size for print of each figure.

Reviewer #3: The authors need to check whether they used the correct Figure numbers in the text (and SI) after the modifications please.

Author: The Figure numbers has been checked after the modifications